# Monitoring the ocean heat content change and the Earth energy imbalance from space altimetry and space gravimetry

Florence Marti[1], Alejandro Blazquez[2], Benoit Meyssignac[2], Michaël Ablain[1], Anne Barnoud[1], Robin Fraudeau[1], Rémi Jugier[1], Jonathan Chenal[2,3], Gilles Larnicol[1], Julia Pfeffer[1], Marco Restano[4], Jérôme Benveniste[5]

[1]MAGELLIUM, Ramonville Saint-Agne, 31520, France
[2]LEGOS, Université de Toulouse, CNES, CNRS, UPS, IRD, 31000, Toulouse, France
[3]Ecole nationale des ponts et chaussées, Marne-la-Vallée, 77420, France
[4]SERCO-ESRIN, Frascati, 44, Italy
[5]ESA/ESRIN, Frascati, 44, Italy

Correspondence to: Florence Marti (florence.marti@magellium.fr)

**Abstract.** The Earth energy imbalance (EEI) at the top of the atmosphere is responsible for the accumulation of heat in the climate system. Monitoring the EEI is therefore necessary to better understand the Earth's warming climate. Measuring the EEI is challenging as it is a globally integrated variable whose variations are small (0.5-1 W m$^{-2}$) compared to the amount of energy entering and leaving the climate system (~ 340 W m$^{-2}$). Since the ocean absorbs more than 90 % of the excess energy stored by the Earth system, estimating the ocean heat content (OHC) change provides an accurate proxy of the EEI. This study provides a space geodetic estimation of the OHC changes at global and regional scales based on the combination of space altimetry and space gravimetry measurements. From this estimate, the global variations in the EEI are derived with realistic estimates of its uncertainty. The mean EEI value is estimated at +0.74±0.22 W m$^{-2}$ (90 % confidence level) between August 2002 and August 2016. Comparisons against estimates based on Argo data and on CERES measurements show good agreement within the error bars of the global mean and the time variations in EEI. Further improvements are needed to reduce uncertainties and to improve the time series especially at interannual time scales. The space geodetic OHC-EEI product (version 2.1) is freely available at https://doi.org/10.24400/527896/a01-2020.003 (Magellium/LEGOS, 2020).

## 1 Introduction

Over the last decades, greenhouse gases and aerosols concentrations have been increasing in the atmosphere, disrupting the balance in the Earth system between incoming and outgoing radiation fluxes. Part of the outgoing longwave radiation being blocked, the system has reemitted less energy towards space than it has received from the Sun (Hansen et al., 2011; Trenberth et al., 2014). This imbalance at the top of the atmosphere, known as the Earth energy imbalance (EEI), is about 0.5-1 W m$^{-2}$ (von Schuckmann et al., 2016). It is challenging to estimate the EEI from top-of-atmosphere radiation fluxes since it is two orders of magnitude smaller than the mean incoming solar radiation (340 W m$^{-2}$) (L'Ecuyer et al., 2015).

Positive values of the EEI indicate that an excess of energy is stored in the climate system. With its high thermal inertia and its large volume, the ocean acts as a buffer, accumulating most of the excess of energy (more than 90 %, e.g. von Schuckmann et al., 2020b) in the form of heat. The other climate reservoirs, the atmosphere, land and cryosphere, play a minor role in the

energy storage at seasonal and longer time scales (von Schuckmann et al., 2020b). As a result, the ocean heat uptake (OHU) prevails in the global energy budget on timescales longer than several months. The global OHU (GOHU) is therefore a good proxy of the EEI variations.

The OHU is positive when heat enters the ocean and negative when heat leaves the ocean. It is the time derivative of the ocean heat content (OHC). OHC change time series may be inferred by different approaches: (1) direct measurement of

temperature/salinity profiles mainly derived from the Argo floats network (von Schuckmann et al., 2020b), (2) re-analysis which combines in situ measurements of temperature/salinity and space measurements of sea level with ocean modelling (Stammer et al., 2016), (3) through the ocean surface net flux from satellite observations (Kato et al., 2018; L'Ecuyer et al., 2015), (4) and the space geodetic approach (introduced in Meyssignac et al., 2019 and this study, see also Hakuba et al., 2021). These methods are complementary, with their own advantages and limitations (Meyssignac et al., 2019). The direct

measurement approach relies on in situ measurements which are unevenly spatially distributed with poor sampling of the deep ocean (below 2000 m depth), marginal seas and below seasonal sea-ice. Re-analyses provide a more complete description of the ocean's state that is consistent with the dynamics of the ocean but are subject to large biases in the polar oceans, spurious drifts in the deep ocean and inaccurate initial conditions that may obfuscate a significant part of the OHC signal related to EEI (Palmer et al., 2017). The ocean net flux approach assesses the radiative and turbulent fluxes from satellite observations to

provide the spatial distribution of net heat fluxes at the ocean surface, but it is tainted with large residuals and uncertainties (Kato et al., 2018; L'Ecuyer et al., 2015). The space geodetic approach aims at measuring the sea level changes due to the thermal expansion and saline contraction of the ocean (also called steric sea level changes) based on differences between the total sea level changes derived from satellite altimetry measurements and the barystatic sea level changes from satellite gravity measurements. This approach offers consistent spatial and temporal sampling of the ocean, with a nearly global coverage of

the oceans, except for the polar regions (above 82°). It also provides OHC change estimates over the entire ocean water column. The EEI shows time variations in response to anthropogenic emissions and natural variability like ocean-atmosphere interactions or volcanic eruptions. The coupled natural variability of the ocean and of the atmosphere leads to monthly to interannual variations of the order of a few W m$^{-2}$ (e.g. Loeb et al., 2018a). Decadal and longer-term variations of the order of a few tenths of W m$^{-2}$ are associated to the anthropogenic and the natural forcing of the climate system (e.g. von Schuckmann

et al., 2016). To evaluate these variations and particularly the small decadal and longer-term response of EEI to anthropogenic or natural forcing, EEI should be estimated with an accuracy better than 0.1 W m$^{-2}$. This is particularly challenging and it requires a fine characterisation of the errors associated with the EEI estimates.

The originality of this study is to provide the OHC change and EEI from space altimetry and space gravimetry with a comprehensive description of the uncertainty. This space geodetic approach has three major advantages: it covers the ocean

down to the bottom, the spatial coverage is nearly global (until 82° poleward) and it is based on a few instruments which

enables an exhaustive description of error sources and a robust propagation of errors from the measurements to the global OHC (GOHC) change estimate. A preliminary estimate of the GOHC 10-year-trend uncertainty of +/-0.32 W m$^{-2}$ (90 % confidence level -CL-) has been published with this approach (Meyssignac et al., 2019). A central objective of this study is to revisit this uncertainty estimate with a realistic and robust uncertainty propagation scheme to enable its computation over any

time span and help reduce uncertainty. First, we provide regional and global estimates of OHC change over the period 2002 to 2016. Second, we rigorously and accurately assess the uncertainty in GOHC change and EEI, propagating the errors from the sea level and ocean mass change estimates and taking into account the time correlations in errors. To reach these objectives, innovative algorithms have been developed. We present them in this paper.

The physical assumptions underlying the estimation of the EEI from space geodetic measurements are introduced in section

2. Section 3 describes the sea level and ocean mass variations, and thermal expansion data used as input for the computation of OHC changes and the EEI over the 15-year period from August 2002 to August 2016 (sect. 4.1). Error propagation and uncertainty calculation are performed independently (sect. 4.2). Results are gathered in sections 5 and 6 for the OHC change and the EEI respectively, including comparisons with estimates mainly based on the in situ Argo network. Conclusions and perspectives for improvement of the EEI record are given in section 7.

In this article, all uncertainties are reported with a 5 %-95 % confidence level interval (also noted 90 % CL).

**2 Physical principle**

In the space geodetic approach, OHC changes are estimated from steric sea level changes, which are due to the thermal expansion and the haline contraction of the ocean column of water. Steric sea level changes are calculated as the difference between total sea level changes and ocean mass changes (e.g. Forget and Ponte, 2015; Meyssignac et al., 2017 and references

there in). It is expressed by the sea level budget equation where the total sea level change ($\Delta SL_{total}$) is the sum of the ocean mass change ($\Delta SL_{mass}$) and the ocean steric sea level change. The latter is composed of two terms, the ocean thermal expansion change ($\Delta SL_{thermosteric}$) and the ocean halosteric change ($\Delta SL_{halosteric}$) following Eq. (1):

$$\Delta SL_{total} = \Delta SL_{\mathrm{mass}} + \Delta SL_{thermosteric} + \Delta SL_{halosteric}. \tag{1}$$

At global scale, the ocean salinity change is negligible (Gregory and Lowe, 2000; Llovel et al., 2019; Gregory et al., 2019), as

it only contributes to about 1 % of the global mean sea level change (Gregory and Lowe, 2000). Therefore Eq. (1). can be simplified, and the global mean thermosteric sea level change ($\Delta GMTSL$) is obtained from the difference between the global mean sea level change ($\Delta GMSL$) and the global mean ocean mass change ($\Delta GMOM$):

$$\Delta GMTSL = \Delta GMSL - \Delta GMOM. \tag{2}$$

Then, the GOHC change ($\Delta GOHC$) is derived by dividing the thermal expansion change by the expansion efficiency of heat (EEH), noted $\varepsilon$ at global scale as in Eq. (3) (see Melet and Meyssignac, 2015 for more details):

$$\Delta GOHC = \frac{\Delta GMTSL}{\varepsilon}. \tag{3}$$

(Note that, at global scale, on multiannual time scales, because the current rate of ocean warming is greater than the interannual variability in GOHC - see for e.g. Cheng et al., 2021 -, $\Delta GOHC$ is always positive and the EEH is always defined and calculable as $\varepsilon = \frac{\Delta GMTSL}{\Delta GOHC}$).

At global scale, on annual and longer time scales, the heat stored by the Earth in response to the EEI is stored essentially in the ocean because the heat capacity of the ocean is much larger than the heat capacity of the rest of the climate system (Palmer and McNeall, 2014; Melet and Meyssignac, 2015). The fraction of energy entering the ocean α is around 0.9. The EEI can now be retrieved from the GOHU, the temporal derivative of GOHC, by dividing it by α the fraction of energy entering the ocean (Eq.(4)). α is set to 0.9, the recent estimate from von Schuckmann et al., 2020b. Beforehand, GOHC change is filtered out to remove the signals related to the intrinsic ocean variability, mostly happening in the mixed layer above the pycnocline. For short time scales (< 2-3 years), this signal does not correspond to any response to global warming (Palmer and McNeall, 2014) and therefore must be removed to infer variations in the EEI:

$$EEI = \frac{GOHU}{\alpha} = 1/\alpha \ \frac{d \ GOHC}{dt}, \tag{4}$$

At regional scale the physical principles are identical as for the global scale except for two differences. First, the regional sea level changes ($\Delta SL$) depend on the salinity changes and thus to derive the regional thermosteric sea level changes ($\Delta TSL$) we need to correct for the regional halosteric sea level changes ($\Delta HSL$) effect following the equation:

$$\Delta TSL = \Delta SL - \Delta OM - \Delta HSL. \tag{5}$$

Second, at a regional scale, it occurs for a water column that the OHC change over the entire column is null while the thermosteric sea level change is not. A typical example is when the heat uptake of a water column above the thermocline is compensated by an equivalent heat loss below the thermocline. In such a case the total heat uptake of the entire water column is by definition zero but the thermosteric sea level change is strictly postive. This is because the expansion of the sea water above the thermocline (which occurs in warmer water) exceeds the contraction below the thermocline (which occurs in colder water). In such a situation the expansion efficiency of heat is not defined and cannot be calculated. A way around this issue is to consider ocean heat content (OHC) (rather than ocean content changes $\Delta OHC$) and thermosteric sea level (TSL) (rather than thermosteric sea level changes $\Delta TSL$) and to define an integrated expansion efficiency of heat (IEEH) **E** as follows:

$$E = \frac{TSL}{OHC}. \tag{6}$$

The IEEH is in m J$^{-1}$ as the EEH. At regional scale, the IEEH is always calculable because the ocean heat content is never null. Thus, the IEEH allows to derive estimates of regional OHU from estimates of the regional thermosteric sea level (TSL) with the following equation:

$$OHU = \frac{d\,OHC}{dt} = \frac{d\,(TSL/\boldsymbol{E})}{dt}.$$ (7)

In this work we use equations 3 and 4 to derive estimates of the GOHU and the EEI. We use equations 6 and 7 to derive estimates of the regional OHU. We verify the consistency of the global and regional estimates of the ocean heat uptake by comparing the global sum of OHU with GOHU (see section 4).

In this study, total sea level change is observed from space with radar altimetry missions (see section 3.1), ocean mass
change is observed from space with the gravimetry missions (see section 3.2) and the global EEH and regional IEEH are estimated from in situ observations of ocean temperature and salinity (see section 3.3). Although EEH and IEEH are derived from in situ data, this approach is called "space geodetic approach" because all dynamic variables are retrieved from satellite remote sensing.

## 3 Data

### 3.1 Sea level

In this study we used sea level daily gridded dataset for the global ocean (Taburet et al., 2019; Legeais et al., in prep) that is distributed by the Copernicus Climate Service (C3S) and contains the sea level anomalies around a mean sea surface above the reference mean sea surface computed over 1993-2012, also referred to as the total sea level change. Data is available over the entire altimetry area from January 1993 onward. They are provided on a daily basis at a spatial resolution of 0.25° x 0.25°.
Thanks to rigorous processing of altimetry measurements based on a two-satellite altimetry constellation, homogeneous altimetry standards applied over time (e.g. geophysical corrections, orbit solutions, etc.) and solid validation activities carried out upstream, C3S sea level products are dedicated to the monitoring of the long-term sea level variations. As C3S sea level grids are not corrected for the global isostatic adjustment (GIA), a correction is applied a posteriori. It is derived from an ensemble mean of regional GIA corrections computed with the ICE-5G model and with various viscosity profiles (27 profiles)
used in Prandi et al., 2021 (Spada and Melini, 2019). The average GIA value over oceans is -0.28 mm yr$^{-1}$ close to the generally accepted value of -0.3 mm yr$^{-1}$ (e.g. WCRP Global Sea Level Budget Group, 2018). An additional correction of -0.1 mm yr$^{-1}$ (GRD) is considered for the deformations of the ocean bottom in response to modern melt of land ice (Frederikse et al., 2017). The description of the errors, and the uncertainties on the long-term stability of the sea level estimate in these products were provided by Ablain et al. (2019) and Prandi et al., (2021) for the global and regional scales respectively. Over the whole
altimetry period (January 1993-December 2020), the GMSL shows a significant rise of +3.52 ± 0.35 mm yr$^{-1}$. Focusing on the period of interest in this study (August 2002-August 2016), the GMSL increase is +3.57 ± 0.40 mm yr$^{-1}$ (AVISO GMSL

indicator). At the regional scale, the sea level rise distribution ranges between 0 and 6 mm yr$^{-1}$, with uncertainties ranging from $\pm 0.8$ to $\pm 1.2$ mm yr$^{-1}$, pointing out that the sea level is rising everywhere over the globe. Recent studies also showed that sea level is accelerating at $0.12 \pm 0.07$ mm yr$^{-2}$ at the global scale (Ablain et al., 2019) and ranges between -1 mm yr$^{-2}$ and +1 mm yr$^{-2}$ at the regional scale (Prandi et al., 2021).

## 3.2 Ocean mass

The Gravity Recovery And Climate Experiment (GRACE) mission, launched in 2002, allowed continuous monitoring of ocean mass change over the study period (Tapley et al., 2004). GRACE was decommissioned in 2017 and its successor GRACE Follow-On (GRACE-FO) was launched in May 2018. This study stands as a proof of concept, demonstrating the capability to deliver space geodetic estimates of the OHC change and EEI and their associated uncertainties. The study period is therefore limited to April 2002-August 2016 when the GRACE data shows the best quality. This restricted period enable to avoid (i) instrumental issues deprecating the quality of the GRACE data at the end of the mission (e.g. Wouters et al., 2014), (ii) the 11-months data gap between GRACE and GRACE-FO, (iii) instrumental issues during the GRACE-FO mission on the accelerometers, (iv) eventual biases between the GRACE and GRACE-FO missions (e.g. Chen et al., 2020; Landerer et al., 2020). Ocean mass variations observed by GRACE are mainly due to freshwater exchanges with the continents (including ice-melting and water cycle) at global scale, and, also, to the ocean circulation at regional scale. However, estimating the rates of global and regional ocean mass change with GRACE data remains a challenging task due to numerous processing choices that can strongly affect the results and lead to a large variety of solutions with significant uncertainty (Uebbing et al., 2019). In this study, we considered the GRACE LEGOS ensemble V1.4 (ftp://ftp.legos.obs-mip.fr/pub/soa/gravimetrie/grace_legos/V1.4/) updated from Blazquez et al. (2018). This ensemble version includes 216 solutions, based on fully normalised spherical harmonic solutions from six different centers and a large variety of choices for post-processing corrections including the corrections of the geocenter motion, the oblateness of the Earth, the atmosphere ocean dealiasing, the filtering of the noise responsible for the characteristic stripes of GRACE gravity data, the leakage correction and the GIA. More details of this update and the appropriate references can be found in the appendix A. This ensemble approach allows a robust estimation of the uncertainties associated with state-of-the-art ocean mass change estimates based on GRACE measurements (see Blazquez et al. (2018) for more details). In addition to spherical harmonics, the ocean mass change can also be estimated from mascon solutions provided by the Jet Propulsion Laboratory (JPL RL06), the Center for Space Research (CSR RL06) and the Goddard Space Flight Center (GSFC RL06). These three mascon solutions use the same post-processing corrections for the geocenter motion (Sun et al., 2016), for the oblateness of the Earth (C20) and the low harmonic degrees (C30) of the gravity field (Loomis et al., 2019), for the dealiasing of the atmosphere and ocean signals (AOB1B RL06 from Dobslaw et al., 2017) and for GIA (ICE6G-D from Peltier et al., 2018). Comparing these three mascons with the subset of the LEGOS ensemble that use the same post-processing corrections lead to similar ocean mass change estimates (see Fig. A1 in appendix A) which confirms the consistency of the mascon solutions with the spherical harmonics solutions and gives confidence in their representation of mass transport. Within the LEGOS ensemble, the subset which uses the mascon post-processing choices shows ocean mass

changes in the upper range of the ensemble estimates. This corroborates the major role of post-processing choices on the estimation of global ocean mass change estimates and stresses the need to quantify the associated uncertainty.

When considering the same mask as the altimetry product, the GMOM trend in the LEGOS ensemble reaches $1.83 \pm 0.21$ mm yr$^{-1}$ for the period from August 2002 to August 2016, in agreement with the state-of-the-art estimates. Regional variations in ocean mass trends are fairly small (up to 3.66 mm yr$^{-1}$) when considering the ensemble mean, except at high latitudes and for

shallow seas, variations in the ocean bottom pressure due to the ocean circulation or changes in the geoid are relatively small compared to the global ocean mass increase (Piecuch and Ponte, 2011; Piecuch et al., 2013).

**3.3 Expansion efficiency of heat (EEH) and integrated expansion efficiency of heat (IEEH)**

The EEH expresses the change in ocean density due to heat uptake. It represents the ratio of the thermosteric sea level change over the heat content change under a given heat uptake. As such it allows estimating changes in OHC from changes in

thermosteric sea level (following Eq. (3)). The EEH can be calculated from known ocean variables (IOC et al., 2010) as the derivative of specific volume with respect to temperature (m$^3$ kg$^{-1}$ °C$^{-1}$) divided by specific heat capacity (J kg$^{-1}$ °C$^{-1}$). The EEH is dependent on temperature, salinity and pressure, it increases with temperature, salinity and pressure (Russell et al., 2000). Thus, integrated over the entire water column the EEH is expected to mainly vary with latitude, together with vertically integrated salt content and temperature. In time, the change in EEH is expected to be negligible over the study period, because

the warming pattern is unlikely to change much at decadal time scales (Russell et al., 2000; Kuhlbrodt and Gregory, 2012). The IEEH is different from the EEH. The IEEH expresses the ratio of the thermosteric sea level over the heat content. As such it allows estimating OHC from thermosteric sea level (following Eq. (6)). The IEEH can be calculated from known ocean variables (IOC et al., 2010) as the specific volume (m$^3$ kg$^{-1}$) divided by the specific enthalpy (J kg$^{-1}$). The IEEH is dependent on temperature, salinity and pressure, it increases with temperature and pressure and decreases with salinity (see Fig. B1 in

appendix B). Note that, because IEEH decreases with salinity while EEH increases with salinity, when integrated over the entire water column, the spatial variations of the IEEH are expected to be different from the spatial variations in EEH.

For the calculation of EEH at global scale, monthly gridded temperature and salinity fields from 11 Argo solutions were used to compute the ratio between GMTSL change and GOHC change. These monthly ratios are averaged over time, then averaged

together to provide a global EEH estimate of $0.145 \pm 0.001$ m YJ$^{-1}$ representative of the 0–2000 m ocean column for the period 2005-2015, excluding marginal seas and areas located above 66° N and 66° S. This regional extent corresponds to the spatial extent that is regularly sampled by the in situ Argo network. The global EEH estimated here is in good agreement with previous estimates of $0.12 \pm 0.01$ m YJ$^{-1}$ (equivalent to 0.52 W m$^{-2}$/mm yr$^{-1}$) representative of the 0–2000 m ocean column over 1955–2010 from in situ observations (Levitus et al., 2012) and $0.15 \pm 0.03$ m YJ$^{-1}$ for the full ocean depth over 1972–2008 (Church

et al. 2011). Its uncertainty is however much smaller because the EEH computation is based on the Argo network that has a precise estimate of ocean temperature and salinity down to 2000 m depth and relies only on effective measurements that were processed homogeneously (eg. interpolated data are excluded, the same horizontal and vertical mask is used). Previous studies

from Levitus et al., (2012) and Church et al., (2011) used an ensemble of temperature and salinity products that covered the whole ocean over the past decades with in-filled data where measurements are lacking. The differences in the in-filled data explain the large uncertainty Levitus et al., (2012) and Church et al., (2011) found in the estimate of the EEH. Here we restricted the study to the region and the time span covered by Argo. Our approach based on recent data products that sample the global ocean provides a more accurate estimate of the EEH which enables to significantly reduce the uncertainties of the GOHC change estimate (see section 4.2. on the error propagation and uncertainty calculation). However, as the sampling of Argo is not fully global (measurements are sparser above 66° latitude and below 2000 m depth) our estimate of the global EEH is likely biased by a few percent. It is likely biased high because the bottom layer, below 2000 m depth, is less salty than upper layers which would result in a slightly lower global EEH estimate if it was accounted for in the computation.

For the calculation of the IEEH at regional scale, monthly gridded temperature and salinity fields from 11 Argo solutions were also used to compute the ratio between local TSL and local OHC. Figure 1 shows the associated spatial grid (3x3 degree) of IEEH estimate (allowing at the same time to visualise its spatial availability). The value of the IEEH for each cell is the temporal mean of the ratio between the local TSL and the local OHC over the period 2005 - 2015. The IEEH grid is applied in this study to calculate OHC at regional scales (see section 4.1. OHC change and EEI calculation) and further derive the regional OHU.

### 3.4 Ancillary data

For comparison purposes, OHC change and EEI are also estimated from in situ ocean temperature and salinity from Argo datasets covering the first 2000 m depth range. We considered the IAP, IFREMER, IPRC, ISHII, EN4, JAMSTEC, NOAA, and SIO datasets. Differences in ocean temperature among these products are due to the different strategies in data editing, temporal and spatial data gap filling and instrument bias corrections (see for e.g. Boyer et al., 2016). All Argo products are post-processed homogeneously in the framework of this study for integration of temperature and salinity to derive the ocean heat content (e.g. one single integration scheme, climatology computed over the same period 2005-2015). Regional OHC change is retrieved relying on the thermodynamic equation of seawater (McDougall and Barker, 2011). Although IAP, IFREMER, ISHII, EN4, and NOAA products extrapolate the temperature and salinity profiles over the whole ocean, the ensemble of Argo-based GOHC change is calculated here after applying the most restrictive Argo geographical mask among Argo products (it corresponds to the Argo mask of the SIO product, see Fig. 1 for the spatial extent of the mask). This approach enables to get consistent and comparable GOHC change from the different Argo products. A deep ocean contribution of heat storage of + 0.07 ± 0.06 W m$^{-2}$ is added for the layers below 2000 m (following (Purkey and Johnson, 2010; Desbruyères et al., 2016). Argo-based EEI estimates are then derived from Argo-based GOHC change with the same method as for the space geodetic approach described in section 4.1. The different Argo products provide heterogeneous uncertainty estimates. Different products consider different sources of uncertainty and none of the products provide a comprehensive estimate of the

uncertainties (see Table 1 in Meyssignac et al., 2019). The absence of a common reference estimate of the uncertainty in Argo gridded temperature products is an issue that has been identified in the climate community. There is currently a community effort that is undertaken in the World Climate Research Program (the GEWEX EEI assessment, see http://gewex-eei.org/) to tackle this problem. This effort should take a few years and the results are not available yet. For the time being uncertainties

on the Argo-based GOHC change and EEI are derived from the ensemble dispersion. This type of uncertainty mainly describes the discrepancy between the various center products involved in the ensemble. It represents the uncertainty associated with different approaches to develop the data quality control and the data processing. It does not comprise any errors related to time and space correlation in temperature measurements or potential systematic temperature and salinity measurement biases among products and potential systematic sampling biases among products. So these uncertainty estimates are likely underestimated.

OHC change estimate is also provided by the Ocean Monitoring Indicator (OMI) from the Copernicus Marine Service (CMEMS) (von Schuckmann et al., 2020a). The yearly indicator is the ensemble mean of 5 GOHC change solutions from reanalyses and optimal interpolations of altimetry data and in situ measurements (including Argo data). The OMI indicator is based on integrated temperature differences along a vertical profile in the ocean, down to 700 m depth, and averaged between 60° S and 60° N. Note that uncertainties on the CMEMS GOHC change are also derived from the ensemble dispersion.

EEI variations are also observed from space by the Clouds and the Earth's Radiant Energy System (CERES) instruments. They enable monitoring the incoming and outgoing radiative fluxes at the top-of-atmosphere. CERES instruments allow retrieving EEI variations (EBAF TOA fluxes, 2019) from weekly to decadal timescales with an uncertainty of $\pm0.1$ W m$^{-2}$ but the time-mean EEI is measured with an accuracy of $\pm3.0$ W m$^{-2}$ due to calibration issues (Loeb et al., 2018b).

Our estimates of OHC change and EEI are compared with OHC change and EEI estimates from Argo, reanalyses and CERES

in sections 5 and 6.

## 4 Data Processing

### 4.1 OHC change and EEI calculation

AA dedicated data processing chain was specifically developed in order to calculate the OHC change and the EEI from space geodetic measurements, following the physical principle described in section 2. Changes in OHC at global and regional scales,

and the EEI are provided in a dedicated product referred to as "MOHeaCAN v2.1" (see section 7).

The first step consists in preprocessing the time series of total sea level and ocean mass change over the specific period. Total sea level and ocean mass change grids are downsampled to a 3x3 degree spatial resolution (~300 km) and averaged on a monthly basis to match the effective spatial and temporal resolutions of GRACE products.

The second step is dedicated to the calculation of the global time series of OHC change and the EEI. GMSL and GMOM time

series are calculated at each time step (monthly) using a weighted average taking into account the sea surface in each cell. The GOHC change is then obtained by taking the difference between the GMSL and GMOM time series (Eq. (2)) and dividing by the global value of EEH coefficient (Eq. (3)). GOHC change is expressed per unit of area (J m$^{-2}$), when divided by the surface

of the Earth at the top of the atmosphere (5.13 $10^{14}$ m²), for a reference height of the top of the atmosphere at 20 km altitude (the same as EBAF, Loeb et al., 2018b). The EEI estimate is then derived from the temporal variations of the GOHC, by

calculating the derivative, i.e. the GOHU, using numerical forward differences and adjusting it to account for energy contributions from other climate reservoirs (Eq. (4)). Beforehand, GOHC change time series is filtered out by applying a low pass filter (Lanczos) with a cut-off period of 3 years in order to remove high-frequency content related to the intrinsic ocean variability (Palmer and McNeall, 2014) and the mesoscale activity that is visible in altimetry but not in gravimetry (described in section 2.).

The last step aims at calculating changes in OHC at regional scales. Monthly steric sea level grids are directly deduced at 3x3 degree spatial resolution from the difference between the collocated sea level and ocean mass grids. Contrary to the global scale, the ocean salinity change cannot be neglected at regional scales (see Eq. (1)), and halosteric contribution to sea level expansion should be removed to retrieve the regional thermal expansion variations of the ocean. Nevertheless, at this stage of the study, the regional OHC change grids are obtained from the steric sea level grids divided by the grid of IEEH coefficients

without accounting for ocean salinity change and therefore should be interpreted carefully. This has no impact on the estimate of the OHC trend over the full period 2005-2015 because the IEEH has been calculated over this period and the salinity effect is thus implicitly counted in the local IEEH coefficients. However, over other periods or smaller periods included within 2005-2015, the local IEEH is expected to be slightly different as the local salinity changes with time and this calculation of the OHC should be considered as an approximation. The approximation is accurate at the level of a few percent because local changes

in salinity are small compared to the total salt content of the water column (according to the Argo record). In this study we have chosen this conservative approach with a constant IEEH because the salinity anomaly data shows important inconsistencies at annual and inter-annual time scales among Argo products (e.g. Ponte et al., 2021). Instead of using low-confidence salinity anomaly data we prefer to assume at this stage a constant IEEH estimated from salinity climatologies that are more reliable. This approach leads to an estimate of regional OHC with a lower uncertainty but the downside is that the

level of confidence in regional OHC is lower. Note that the GOHC change can also be deduced from the regional OHC grids by computing regional OHC anomalies and summing all cells weighted by their area. We checked this approach and found that it leads to similar estimates of GOHC change and GOHU to those of the global approach described before in equation 3.

## 4.2 Error propagation and uncertainty calculation at global scale

One of our main objectives is to provide the uncertainty associated with the OHC change and EEI estimates. In this study the

error propagation is performed only at global scale. It is much more complex to propagate the uncertainty at regional scales because it requires to describe the spatial correlation of the errors in satellite altimetry and space gravimetry data which is not a simple task. At this time estimates of these errors are not available in the literature but this work is currently ongoing and should be the subject of further publications in the coming years. Meanwhile we focus on the uncertainty at global scale. A rigorous approach is proposed here, providing the variance-covariance matrix ($\Sigma$) of the errors for the GOHC change and EEI

time series at global scale. To obtain the $\Sigma$ matrices of the GOHC change and EEI time series, errors must be propagated from

the GMSL and GMOM monthly time series as represented in Fig. 2. The first step consists of estimating the variance-covariance matrices for the sea level ($\Sigma_{GMSL}$) and the ocean mass ($\Sigma_{GMOM}$) time series.

$\Sigma_{GMSL}$ is inferred from the GMSL error budget of Ablain et al. (2019) over the period 2002-2016. In short, the elementary variance-covariance matrices ($\Sigma_{error_i}$) corresponding to each error described in the GMSL error budget (Ablain et al., 2019) are first calculated independently of each other. Each matrix is calculated from a large number of random draws ($> 1000$) of simulated error signals whose correlation is modelled. Their shape depends on the type of errors prescribed, which can be of several kinds: jumps, time-correlated errors, long-term drifts. Assuming errors are independent, $\Sigma_{GMSL}$ is given by the sum of all $\Sigma_{error_i}$ (see Ablain et al., 2019 for the details of the calculation).

For the calculation of $\Sigma_{GMOM}$ we use an ensemble approach where the ensemble of GMOM time series ($X_i$) is directly used to calculate the covariance between each time series:

$$\Sigma_{GMOM}(i,j) = cov(X_i, X_j) = E[(X_i - E[X_i])(X_j - E[X_j])], \tag{8}$$

where E is the mean operator. This approach is reliable when GMOM ensembles are large enough, so that the dispersion between the members of the ensemble adequately represents the GMOM uncertainties. We use this approach with the LEGOS ensemble of 216 ocean mass solutions but we can not apply it with the ensemble of mascon solutions which has only 3 distinct members. For the mascon ensemble, the uncertainty is simply computed as the standard deviation between the three solutions.

The second step consists in calculating the variance-covariance matrices for the GMTSL time series ($\Sigma_{GMTSL}$). The GMTSL is obtained by calculating the differences between the GMSL and the GMOM. We consider the errors in GMSL independent from the errors in GMOM and estimate $\Sigma_{GMTSL}$ as the sum of $\Sigma_{GMSL}$ and $\Sigma_{GMOM}$. Note that this assumption is not verified in reality as some errors are correlated between GMSL and GMOM like the errors related to the GIA correction and the error associated with the positioning of the reference system (in particular to the geocenter position). But the amplitude of these errors is very different in altimetry and space gravimetry. While the error in GIA correction and in the geocenter position are important in space gravimetry (see for e.g. Uebbing et al., 2019; Blazquez et al., 2018), their effect on satellite altimetry is small (see for e.g. Ablain et al., 2019 and reference there in). Thus, on the overall, the correlation in satellite altimetry and space gravimetry of the GIA and the geocenter correction errors is expected to be low and we neglect it here.

In the third step we propagate the errors in the calculation of the GOHC change. As the GOHC change is derived from the GMTSL by dividing it by the global coefficient of EEH $\varepsilon$, the uncertainty on $\varepsilon$ ($e_\varepsilon$) has also to be considered:

$$GOHC(t) = \frac{GMTSL(t) \pm e_{GMTSL}(t)}{\varepsilon \pm e_\varepsilon}, \tag{9}$$

The error propagation for the division of the two uncorrelated variables GMTSL(t) and $\varepsilon$ with a respective uncertainty $e_{GMTSL}(t)$ and $e_\varepsilon$ leads to the following form for the variance-covariance matrix of GOHC change time series ($\Sigma_{GOHC}$) (e.g. see Taylor, 1997, eq. (3.8)):

$$\Sigma_{GOHC} = \frac{1}{\varepsilon^2}\Sigma_{GMTSL} + \left(\frac{\varepsilon}{e_\varepsilon}\right)^2 GOHC * GOHC^t, \tag{10}$$

This equation shows that GOHC errors depend on the uncertainty $e_\epsilon$ but also on the value of $\varepsilon$.

The last step is the propagation of errors in the EEI, obtained after filtering and deriving the GOHC with respect to time and adjusting it with $\alpha$ the fraction of energy entering the ocean. These complex operations do not allow to express the errors of

the EEI with a literal expression as for the GOHC change (Eq. (10)). An empirical approach is then proposed to first derive the variance-covariance matrix of GOHU time series ($\Sigma_{GOHU}$). It firstly consists in generating a set of GOHC errors time series ($e_k$) whose variance-covariance matrix is $\Sigma_{GOHC}$. They are obtained by the product of the Cholesky decomposition of $\Sigma_{GOHC}$ ($\Sigma_{GOHC} = AA^t$), and a random vector ($R_k$) following a Gaussian vector of mean 0 and covariance matrix the identity:

$$e_k = AR_k^t, \tag{11}$$

Each $e_k$ is then filtered by a low-pass filter at 3 years to provide a set of GOHU errors time series from which the variance-covariance matrix $\Sigma_{GOHU}$ is easily inferred (see Eq. (8)). Finally, $\Sigma_{EEI}$ is obtained simply from $\Sigma_{GOHU}$ taking into account the $\alpha$ fraction of energy stored in the ocean but neglecting any of its errors:

$$\Sigma_{EEI} = \frac{1}{\alpha^2}\Sigma_{GOHU}, \tag{12}$$

Once variance-covariance matrices are known, the statistical parameters (e.g. trend, acceleration) can be fit at any time-spans

from a linear regression model ($y = X\beta + \epsilon$) applying an Ordinary Least Square (OLS) approach, where the estimator of $\beta$ with the OLS, noted $\hat{\beta}$, is:

$$\hat{\beta} \sim (X^tX)^{-1}X^ty, \tag{13}$$

and where the distribution of the estimator $\hat{\beta}$ takes into account $\Sigma$ and follows a normal law:

$$\hat{\beta} = N(\beta, (X^tX)^{-1}(X^t\Sigma X)(X^tX)^{-1}), \tag{14}$$

This mathematical formalism was fully described in Ablain et al. (2019) to estimate the uncertainties of the GMSL trend and acceleration. It is applied in this study to derive the realistic uncertainties of GOHC and EEI trends. The uncertainty envelope can also be derived from the square root of the diagonal terms of $\Sigma$.

## 5 Ocean heat content change: results & comparison

### 5.1 Global and regional OHC change

The GOHC trend is +0.70 ±0.20 W m$^{-2}$ for the period from August 2002 to August 2016 (Fig. 3a). It indicates the rate at which oceans accumulate heat and gives an estimate of the average GOHU. This value is significant when compared to its uncertainty

of $\pm 0.20$ W m$^{-2}$. In this trend uncertainty, the contribution from satellite altimetry uncertainty is higher than the contribution from space gravimetry uncertainty (see Table C1 in appendix C). The GMSL error budget provided by Ablain et al. (2019) is by construction comprehensive and conservative (all choices are conservative in particular the representation of the error in wet tropospheric correction and its time correlation are probably slightly overestimated) and leads to GMSL errors that are likely slightly overestimated. In addition, the total GMSL errors have been validated against independent measurements from tide gauges (e.g. Watson et al., 2015) so there is high confidence that the 90 % CL uncertainty in GMSL used here is an upper bound of the real uncertainty in GMSL. GMOM errors are deduced from an ensemble of GRACE solutions (update of Blazquez et al., 2018) accounting for all known sources of errors including instrumental errors (e.g. taken into account using solutions from different centers) and post-processing choices (e.g. geocenter, oblateness, filter, GIA). Although we are confident the current state-of-the-art post processings used in the ensemble of solutions provide a reliable coverage of the real associated uncertainty, we can not rule out the possibility that the resulting GMOM uncertainty is slightly underestimated because of some unknown small undetected systematic bias among state-of-the-art post processing. Another issue is that there is no validation of GMOM against independent data available yet. The global freshwater budget offers a potential approach to validate the GMOM estimates against independent estimates derived from the sea ice volume changes and the ocean global salinity estimates (e.g. Munk, 2003). But the first results show that estimates of the global ocean salinity are not accurate enough to provide an efficient validation (Llovel et al., 2019). For these reasons, we have a smaller confidence in the GMOM uncertainty estimate than in the GMSL uncertainty estimate leading to a confidence in our GOHC change uncertainty estimate that is between medium and high. Note that compared to previous estimates in Meyssignac et al. (2019) the uncertainty in GOHC change is reduced here. This is essentially due to the updated estimate of the global EEH coefficient with Argo data that leads to a smaller uncertainty than the estimate of Levitus et al. (2012) used in Meyssignac et al. (2019) (see section 3.3). Regional OHC trends for the period from August 2002 to August 2016 are generally positive ranging from -1 to +2 10$^{-3}$ W m$^{-2}$ (Fig. 4). As the OHC is an integrative variable, it depends on the area considered in the computation. In this case the difference between the surface considered in the GOHC change and the surface considered in regional OHC change is of the order of 2 10$^{-4}$, explaining the difference of 3 orders of magnitude between the typical GOHC and the typical regional OHC changes. The spatial patterns depicted by the GOHC trends are highly correlated to climate mode fingerprints retrieved for example in steric anomalies (eg. Pfeffer et al., 2018). These include for instance the Pacific Decadal Oscillation, dividing the North Pacific along a typical northeast - southwest chevron pattern (e.g. Mantua and Hare, 2002), and the El Niño–Southern Oscillation (e.g. Enfield and Mayer, 1997), consisting in a typical west-east oscillation of the temperature in the tropical and South Pacific. The spatial patterns observed in the North Atlantic are likely related to the warming of the Gulf Stream in the Northeast Atlantic and to the cooling of the Atlantic Meridional Overturning Circulation (AMOC) bringing warm waters from the tropical Atlantic to the Northwest Atlantic (e.g. Ruiz-Barradas et al., 2018). The positive anomaly in the Indian Ocean is likely related to the warm pool, recording higher temperature increase during the last decades than the global ocean (e.g. Rao et al., 2012; Weller et al., 2016; Lee et al., 2015).

## 5.2 Comparison with estimates based on in situ temperature profiles

To evaluate our GOHC change estimate, we compare it with estimates over the period 2005-2015. The processing of the Argo gridded ocean in situ temperature products into GOHC change time series is described in section 3.4. The comparison is restricted to the period January 2005-December 2015, because the coverage of the Argo network becomes nearly global only after 2005 and because afterwards issues in the Argo salinity products lead to artefacts in the salinity climatology and further in the GOHC change products. Over 2005-2015, the space geodetic GOHC trend of $+0.71 \pm 0.23$ W m$^{-2}$ is in agreement within the uncertainties with the Argo-based GOHC trend of $+0.59 \pm 0.13$ W m$^{-2}$ and also with the CMEMS GOHC trend of $+0.60 \pm 0.25$ W m$^{-2}$ (Table 1).

As an indication, the average GOHC trend deduced from another combination of altimetry and gravity measurements has been also calculated using three GRACE mascon solutions (see Table 1). A low value of 0.56 W m$^{-2}$ is obtained for the 2005-2015 time period, but it is still consistent with the MOHeaCAN product as it is in the uncertainty range of the GOHC trend estimated from the MOHeaCAN product ($+0.71 \pm 0.23$ W m$^{-2}$). To check more precisely the consistency between the mascon-based estimate of the GOHC trend and the MOHeaCAN estimate we re-estimate the MOHeaCAN GOHC trend over 2005-2015 using only the sub-ensemble of GRACE spherical harmonic solutions that is based on the same post-processing choices as the mascon solutions. In this case we find a result ($+0.61 \pm 0.18$ W m$^{-2}$) that is closer by less than 0.05 W m$^{-2}$ to the mascon-based estimate. This precise consistency at the level of 0.05 W m$^{-2}$ gives confidence in our estimate. The residual difference could be due to sources of errors that were omitted in the calculation of the spherical harmonic ensemble, such as incomplete leakage errors or differences in the regularisation process of the mascon solutions and the spherical solutions.

The space geodetic GOHC interannual variations of 5 10$^7$ J m$^{-2}$ are presented in Fig. 5. We find the interannual variations in GOHC change in agreement with Argo-based estimates for time scales greater than 3 years, low during the period from 2006 to 2011, and high during the period from 2011 to 2015 (Fig. 5). At shorter time scales (lower than 3 years), variations in GOHC change are poorly correlated. At these time scales, part of the signal is due to the internal variability of climate (e.g. ENSO) that may not be detected in the same way by both space geodetic and Argo-based estimates because of their different time and space resolution. In addition, GOHC variations depicted by all datasets suffer from a lack of accuracy at these time scales to analyse any differences in a significant way (see the large uncertainty envelope at sub annual time scales shown in Fig. 5).

At regional scale, over the period 2005-2015, space geodetic and Argo-based OHC trends are similar (Fig. 6). Overall there is a fairly good spatial coherence of the observed spatial structures as in the Equatorial Pacific Ocean and in the Northern Atlantic but the amplitude of the signals is systematically higher in the space geodetic OHC trend. In addition some discrepancies are observed in Indian Ocean where space geodetic OHC trends are about two times the Argo-based estimates. Although input data are similar, the OHC trends based on the various Argo datasets also show differences at regional scales up to 2.6 10$^{-3}$ W m$^{-2}$ among different Argo products. This is the same order of magnitude as the difference with the regional MOHeaCAN trends (Fig. 6). These analyses on a regional scale provide insights on the regional structure of the signal. They remain preliminary

and present several limitations. On the one hand, the contribution of the regional halosteric signal is not taken into account here in the calculation of the space geodetic OHC change. Ocean salinity change may have a significant impact in some local regions (as in the southeast Indian Ocean (Llovel and Lee, 2015), in the northwest Indian Ocean or close to the Arctic ocean). On the other hand, the regional contribution of the deep ocean in the Argo data (restricted to 0-2000m) is not considered. These limitations will be the subject of future work and may lead to a better agreement between the OHC trends observed by space geodetic data and Argo data.

## 6 Earth Energy Imbalance: results & comparison

The space geodetic approach provides the mean EEI estimate and also the temporal evolution of the EEI over the 15-year period from August 2002 to August 2016 (Fig. 3). The mean EEI of $+0.74 \pm 0.22$ W m$^{-2}$ is obtained from the GOHC trend corrected to account for the energy uptake from land, cryosphere and atmosphere. This mean EEI value represents an enormous amount of energy when it is integrated over the entire Earth's surface at the top of the atmosphere. It represents a total energy uptake of the Earth of about 350 TW (i.e. about 1000 times the power of the world's nuclear power plants). Our EEI estimate indicates a positive trend of $0.02 \pm 0.05$ W m$^{-2}$ yr$^{-1}$, representing a non-significant acceleration of the energy uptake by the ocean over 2002-2016 (see also Table C1 in appendix C). Longer time series or more accurate data are needed to analyse this acceleration. Our EEI estimate also shows large interannual variations of EEI from -0.5 to 2.0 W m$^{-2}$ (Fig. 3) between 2002 and 2016 that are due to climate-change variations of GOHC change or to internal variability. Further studies are needed to determine the causes for these variations. At 3-year time scales the uncertainty of our EEI estimate varies from 0.8 to 1.0 W m$^{-2}$. These uncertainties are too high to enable the monitoring of the EEI response to anthropogenic or natural forcing that require an accuracy below 0.1 W m$^{-2}$ (e.g. Meyssignac et al., 2019). Lower uncertainties would be necessary to explore the EEI signal at shorter time scales.

A recent study applying the geodetic approach as well shows a value of $+0.77 \pm 0.27$ W m$^{-2}$ over the period 2005-2015 (Hakuba et al., 2021). This result agrees very well with ours ($+0.77 \pm 0.24$ W m$^{-2}$) despite significant differences in the input data, in particular the EEH and the ocean mass.

Our space geodetic EEI is also compared at interannual time scales with Argo-based and CERES-based EEI estimates (Fig. 7). Signals lower than 3 years are filtered out in all EEI time series. EEI means and trends are also removed beforehand from each dataset to compare EEI variations at interannual scales.

The interannual signals are better correlated between the time series from space geodetic and CERES data than with the Argo-based data. Although the amplitude of the space geodetic EEI signal is slightly higher (up to 0.8 W m$^{-2}$), they appear to be fairly well phased between 2006 and 2013 (same phase within a few months). In contrast, the Argo-based EEI have similar amplitudes to those of CERES, but are mostly out of phase. The short time period of the in situ data in particular limits the analysis of these signals. To date, the origin of the discrepancies between these different EEI estimates remains under investigation. They are all impacted by internal variability, in particular ENSO (e.g. mid-2007-mid 2009 (Loeb et al., 2012),

2011) and the high frequency signals (monthly to biannual). Regional signature of the internal variability may not be the same in the different observing systems (owing to their different spatial and temporal resolution) leading to discrepancies in EEI estimates. Observing systems with incomplete coverage may miss some signals at specific spatial and temporal scales that could have a major impact on the global estimate. Another source of discrepancy among EEI estimates is that we assumed for the geodetic approach and the in situ approach that 90 % of the excess of energy due to EEI is captured by the ocean. While

this assumption is reasonable at biannual and longer time scales (Palmer and McNeall, 2014), it is probably not true at smaller time scales when atmosphere and to a smaller extent land and cryosphere exchange larger portions of energy with the ocean. This too simple assumption may explain some discrepancies between the CERES estimate on one side and the geodetic and the in situ estimates on the other side.

## 7 Data availability

Changes in OHC at global and regional scales, and the EEI are gathered in the "climate indicators from space product", or "MOHeaCAN" product v2.1, available online at https://doi.org/10.24400/527896/a01-2020.003 with the complete associated documentation (product user manual and algorithm theoretical basis document).

## 8 Conclusions and outlook

This study provides the first space geodetic estimate with a rigorous uncertainty propagation algorithm of the Earth energy

imbalance and changes in ocean heat content at global scale. It is based on the assumption that monitoring heat accumulation in the ocean, with a combination of satellite altimetry and gravimetry measurements, is representative of the vast majority (~ 90 %) of the energy imbalance observed at the top of the atmosphere. The mean value of the EEI derived from this space geodetic approach over the period August 2002 to August 2016 is $+0.74 \pm 0.22$ W m$^{-2}$. This figure is fully in agreement with data based on in situ measurements (Argo network) within the confidence level of the uncertainty. Furthermore, although this

is a preliminary calculation, the OHC change is also calculated for the first time at regional scale thanks to a set of expansion efficiency of heat coefficients estimated from in situ Argo data. The spatial patterns retrieved in the OHC trends look similar to climate mode fingerprints observed in steric anomalies (e.g. Pfeffer et al., 2018). They also correlate well with regional OHC trends derived from in situ Argo data, despite known limitations in these regional estimates (e.g. deep ocean in Argo data, salinity ocean change not corrected in altimetry and gravimetry approach).

The rigorous uncertainty estimate proposed here has still a few limitations. It does not account for the loss of spatial coverage imposed by the Argo geographical mask in the computation of the expansion efficiency of heat. It does not include either the errors related to the estimation of the global EEH value over the first 2000 m depth only (i.e. the effect of the deep ocean on the EEH value is neglected). Furthermore, no error on the fraction of energy entering the ocean $\alpha$ is included in the EEI uncertainty. Finally, the approach depends on the knowledge of the GMSL and GMOM error budget. These error budgets can

be improved further. In particular, an effort must be made to better describe the errors in spatial gravity measurements, especially to include the uncertainties related to the differences in the harmonic and mascon approaches in the error budget. The consistency between the processing of altimetry and gravimetry data could still be improved for instance by homogenising the GIA datasets used to correct the gravimetry signals and the sea level from altimetry. Also atmospheric effects should be harmonised. Indeed, altimetry data are currently processed with the dynamical atmospheric correction (Carrère and Lyard, 2003) while only the inverse barometer correction is applied in gravimetry processing (Blazquez et al., 2018). Another area for improvement is the extension of the spatial and temporal scales of the OHC change estimation. While altimetry and GRACE data are available together since August 2002, the datasets provided in this study are limited in time (August 2002-August 2016) and space (Argo mask) as the objective of this study is to demonstrate the feasibility of such an approach (proof of concept) using reliable GRACE measurements and EEH/IEEH data over the Argo geographical mask. However, in the future, the OHC change and EEI time series could be extended in time using the GRACE-FO data already available from August 2018. This requires managing issues related to the 11-month gap between GRACE and GRACE-FO data (July 2017-June 2018) and the degradation of GRACE data quality after August 2016. The OHC change could also be estimated outside the current Argo mask by extrapolating the EEH coefficient grid to the full ocean using ocean reanalyses. At this stage of the study, OHC changes and EEI are retrieved in a conservative way. Altimetry and gravimetry grids are resampled with a 3x3 degree spatial resolution and GOHC time series is filtered at 3 years with the aim of mitigating the high frequencies impact from the input geodetic datasets and reducing signals related to internal variability on the EEI. Additional studies are necessary to better apprehend how geodetic data can be combined both on temporal and spatial dimensions so as to investigate regional OHC changes.

This study emphasises that the synergy between spatial data (altimetry and gravity) and in situ data (Argo Network) is essential to obtain accurate estimates of OHC change. The former contributes to observing the total OHC variations over the entire water column and with a very good spatial and temporal resolution since 2002, while the latter provides a quasi-global coverage since 2005 and allows access to the vertical structure of the thermal expansion of the ocean down to 2000 m depth. The capacity of both observing systems to provide independent estimates of the EEI since 2005 is absolutely essential. By pointing to discrepancies among different EEI estimates from different observing systems, intercomparisons foster further development to understand the causes for discrepancies. As we understand these discrepancies, the different estimates will improve and we can expect significantly more precise and more robust estimates of the EEI in the coming decade. It is crucial that the space geodetic observing system and the Argo network continue the monitoring and improve their coverage and accuracy in the years to come to support this effort.

**Appendix A**

The GRACE LEGOS ensemble V1.4:

GRACE LEGOS V1.4 is an ensemble of 216 global water mass transfer solutions derived from GRACE and GRACE-FO mission covering the period from August 2002 to December 2020 at monthly time scale and with a spatial resolution of 1 degree. The total amount of water remains constant from one month to another for each solution. The ensemble is based on L2 spherical harmonic solutions from 6 different centers: COST-G RL1.2, CNES RL5.0, CSR RL06, GFZ RL06, JPL RL06, and

TUGRAZ ITSG2018. Atmosphere and ocean dealiasing models are restored using AOD1B RL06 (Dobslaw et al., 2017) except for the CNES solution where ERA interim and TUGO models where used. Ocean dealiasing model is restored and C0 coefficients are corrected in the spherical harmonics to compensate for the total amount of water vapor in the atmosphere expressed in C0 GAA (Chen et al, 2019). The ensemble includes also a large variety of choices for post-processing corrections including: 3 geocenter motion (Lemoine J-M, Reinquin F., 2017; Uebbing et al., 2019; Sun et al., 2016) , 3 oblateness of the

Earth (Cheng et al., 2013; Lemoine J-M, Reinquin F., 2017; Loomis et al., 2019), 2 GIA correction (ICE6G-D (Peltier et al., 2018) and Caron et al. (2018). In order to reduce the anisotropic noise DDK filters are applied to the L2 solutions, including DDK5 and DDK6 (Kusche et al., 2009) except for the CNES solution where a truncated single value decomposition scheme is used for the inversion instead of a classical Cholesky inversion. This method reduces the noise drastically but on the other hand the coefficients of high degree where information is scarce are normalized to the mean coefficients (Lemoine et al., 2016).

Solid Earth displacement due to the largest Earthquakes (Sumatra 2004 and 2012, Tohoku-Oki 2010 and Chili 2010) are corrected following (Tang et al., 2020). Moreover, a new method to convert from spherical harmonics to equivalent water height is applied (Blazquez et al. in prep). This method consists in using high spatial a priori solutions to reduce leakage and Gibbs effects. The spherical harmonics solution is separated in the a priori part using external data as land/ocean masks, glacier mass trends (Hugonnet et al., 2021) and lake volume change (Crétaux et al., 2016) and the rest of the harmonics solution which

contains less signal and must be filtered.

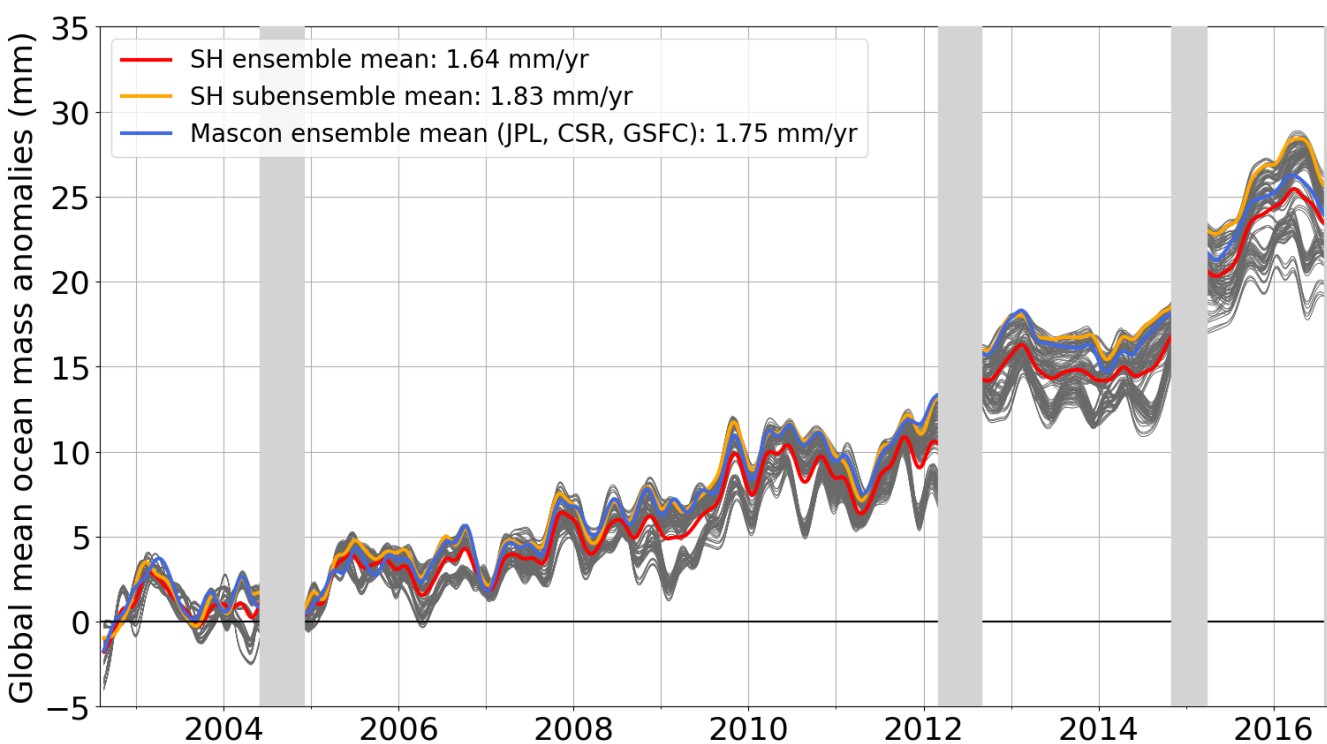

**Figure A1: Comparison of global mean ocean mass changes from satellite gravimetry based on spherical harmonics solutions (LEGOS ensemble V1.4, in grey) and mascon solutions over August 2002-August 2016 for the global ocean. The mean of the full spherical harmonic ensemble is shown in red. The mean of the spherical harmonic ensemble subset consistent with mascons is shown in orange. The mean of the three mascon solutions considered in this study is shown in blue.**

**Appendix B**

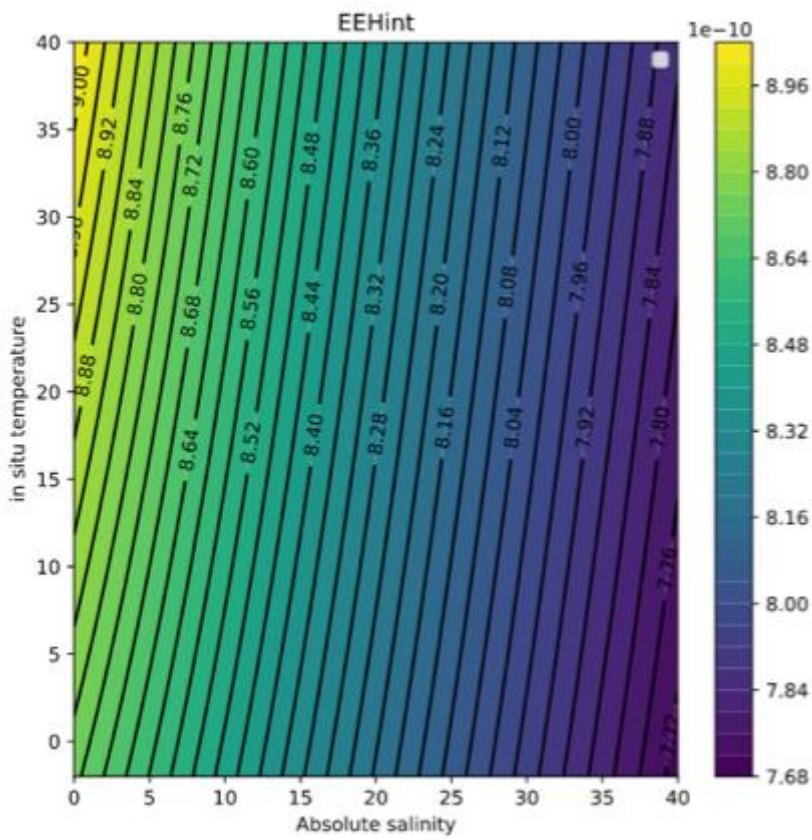


**Figure B1: Integrated expansion efficiency of heat (IEEH) dependence on in situ temperature and absolute salinity at 1 atm in mm/J-1.**

**Table C1. Global mean sea level change components, global ocean heat content changes and Earth energy imbalance - trend/mean values and associated uncertainties as estimated from the various datasets depicted in this paper. Uncertainties are given within a 90 % CL.**

|  | Period | |
| :---: | :---: | :---: |
|  | **8/2002-8/2016** | **1/2005-12/2015** |
| Geocentric sea level change (mm yr$^{-1}$) | +3.11 ± 0.41 | +3.09 ± 0.50 |
| Sea level change (after GIA and GRD corrections) (mm yr$^{-1}$) | +3.49 ± 0.43 | +3.47 ± 0.51 |
| Ocean mass change (mm yr$^{-1}$) | +1.83 ± 0.21 | +1.80 ± 0.21 |
| Steric sea level change (mm yr$^{-1}$) | +1.66 ± 0.48 | +1.67 ± 0.54 |
| Ocean heat content change (W m$^{-2}$) | +0.70 ± 0.20 | +0.71 ± 0.23 |
| Earth energy imbalance (W m$^{-2}$; W m$^{-2}$ yr$^{-1}$) | +0.74 ± 0.22 (mean) +0.02 ± 0.05 (trend) | +0.77 ± 0.24 (mean) +0.08 ± 0.09 (trend) |

**Author contributions**

FM and MA led and designed the manuscript, which was edited by BM, AB and JP. AB and JP focused on the part related to gravimetry observations. FM, RF, RJ and AB developed the processing tools and performed the computations. JC contributed to the gravimetry LEGOS ensemble. BM and MA led and designed the study. GL, MR, and JB supervised the study. All the authors participated in the discussions and revision of the manuscript.

**Competing interests**

The authors declare that they have no conflict of interest.

## Acknowledgments

This work has been supported by ESA in the framework of the MOHeaCAN project (Monitoring Ocean Heat Content and Earth Energy ImbalANce from Space): eo4society.esa.int/projects/moheacan/. This work is also supported by the CNES for the dissemination of the products. We would like to thank in particular Françoise Mertz and Caroline Mercier for making the data available on the ODATIS portal and AVISO. Julia Pfeffer and Anne Barnoud are supported by the European Research Council (ERC) under the European Union's Horizon 2020 research and innovation program (GRACEFUL Synergy Grant agreement No 855677). Jonathan Chenal is grateful to the French Ministry of Ecological Transition for its funding.

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

**Table 1. Global ocean heat content (GOHC) trend and associated uncertainties as estimated from the various datasets depicted in this paper. Uncertainties are given within a 90 % CL.**

| Data type | Source | Spatial coverage (a) Temporal sampling (b) Depth range (c) | GOHC trend (W m⁻²) | |
|---|---|---|---|---|
| | | | 1/2005-12/2015 | 8/2002-8/2016 |
| Temperature and salinity profiles from Argo network | Ensemble of OHC change solutions provided by several international groups[1] | (a) Argo mask (Fig.1) (b) Monthly sampling (c) 0-2000 m + deep ocean contribution of +0.07 W m⁻² | +0.59 ± 0.13[2] | Not available |
| Combination of in situ data (Argo network) and reanalyses | Ensemble of OHC change solutions from CMEMS (Ocean Monitoring Indicator) | (a) Global 60° S-60° N (b) Annual sampling (c) 0-700 m | +0.60 ± 0.25[3] | +0.60 ± 0.25[3] (2003-2016) |
| Space geodetic data | Sea level grids from C3S | Ensemble mean of 216 solutions based on spherical harmonic approach (detailed in this paper) — (a) Argo mask (Fig.1) (b) Monthly sampling (c) 0-bottom | +0.71 ± 0.23 | +0.70 ± 0.20 |
| | | Ensemble mean of 3 solutions based on mascon approach (JPL, CSR, GSFC) | +0.56 ± 0.21[4] | +0.57 ± 0.18[5] |

[1] List of Argo international groups:

EN4 data set from the Met Office Hadley Centre (Good et al., 2013), including MBT and XBT data corrected by Gouretski and Reseghetti (2010) and Levitus et al. (2012),

IAP (Institute of Atmospheric Physics of the Chinese Academy of Sciences), including MBT and XBT data corrected by Gouretski and Reseghetti (2010) and Levitus et al. (2012),

IPRC (combined to altimetry data),

IFREMER (Gaillard et al., 2016; Kolodziejczyk et al., 2017),

Ishii et al. (2017),

JAMSTEC (Japan Agency for Marine-Earth Science and Technology) MILA GPV (Mixed Layer data set of Argo, Grid Point Value) product data set (Hosoda et al., 2010),

NOAA (National Oceanic and Atmospheric Administration) data (Huang et al., 2017),

SIO (Scripps Institution of Oceanography) climatology monthly gridded 1°x1° data (Roemmich and Gilson, 2009).

[2] Uncertainty given by the dispersion of the ensemble and uncertainty on deep ocean contribution

[3] Uncertainty given by the dispersion of the ensemble

[4] Uncertainty derived from the approach described in this study (gravimetry data uncertainty is simply computed as the standard deviation between the 3 mascon solutions). GOHC trends obtained with each mascon dataset - JPL: 0.60 W m$^{-2}$, CSR: 0.55 W m$^{-2}$, GSFC: 0.54 W m$^{-2}$.

[5] Uncertainty derived from the approach described in this study (gravimetry data uncertainty is simply computed as the

standard deviation between the 3 mascon solutions). GOHC trends obtained with each mascon dataset - JPL: 0.61 W m$^{-2}$, CSR: 0.56 W m$^{-2}$, GSFC: 0.54 W m$^{-2}$.



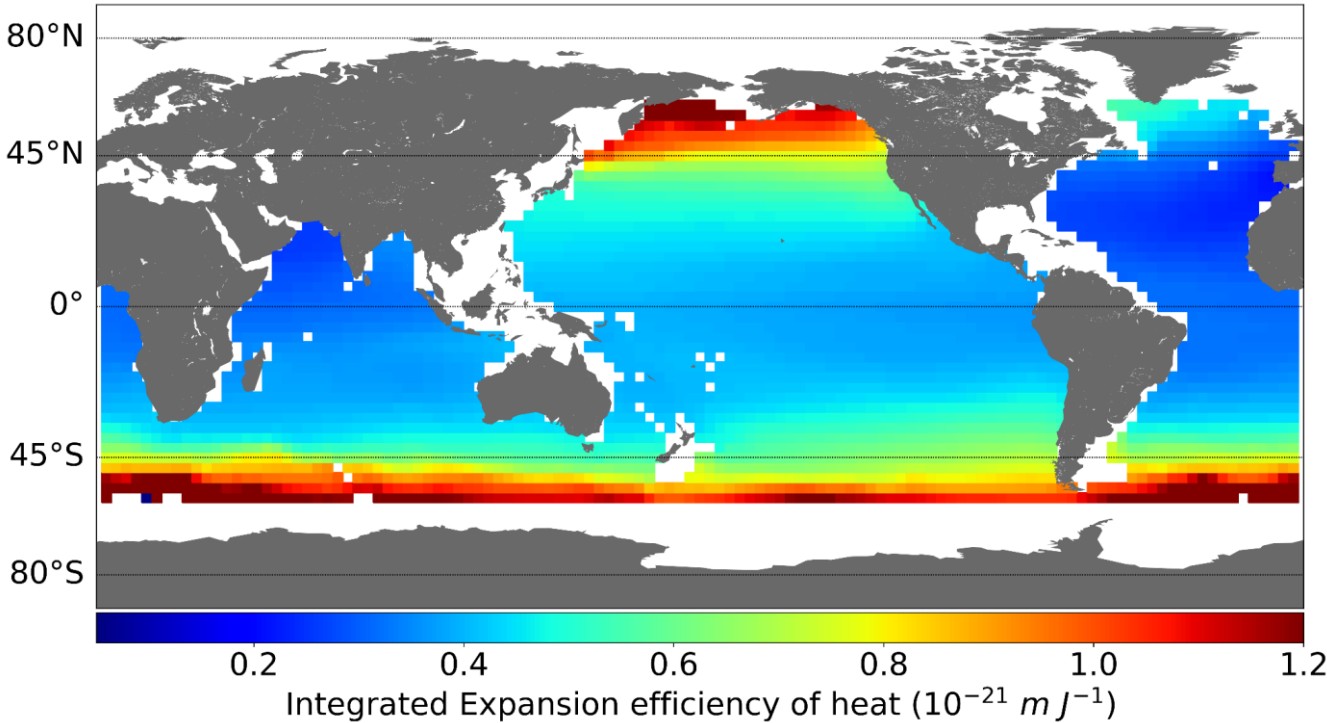

**Figure 1: Integrated Expansion Efficiency of Heat (EEH) coefficients (m J⁻¹) at regional scale (3x3 degrees). See text.**

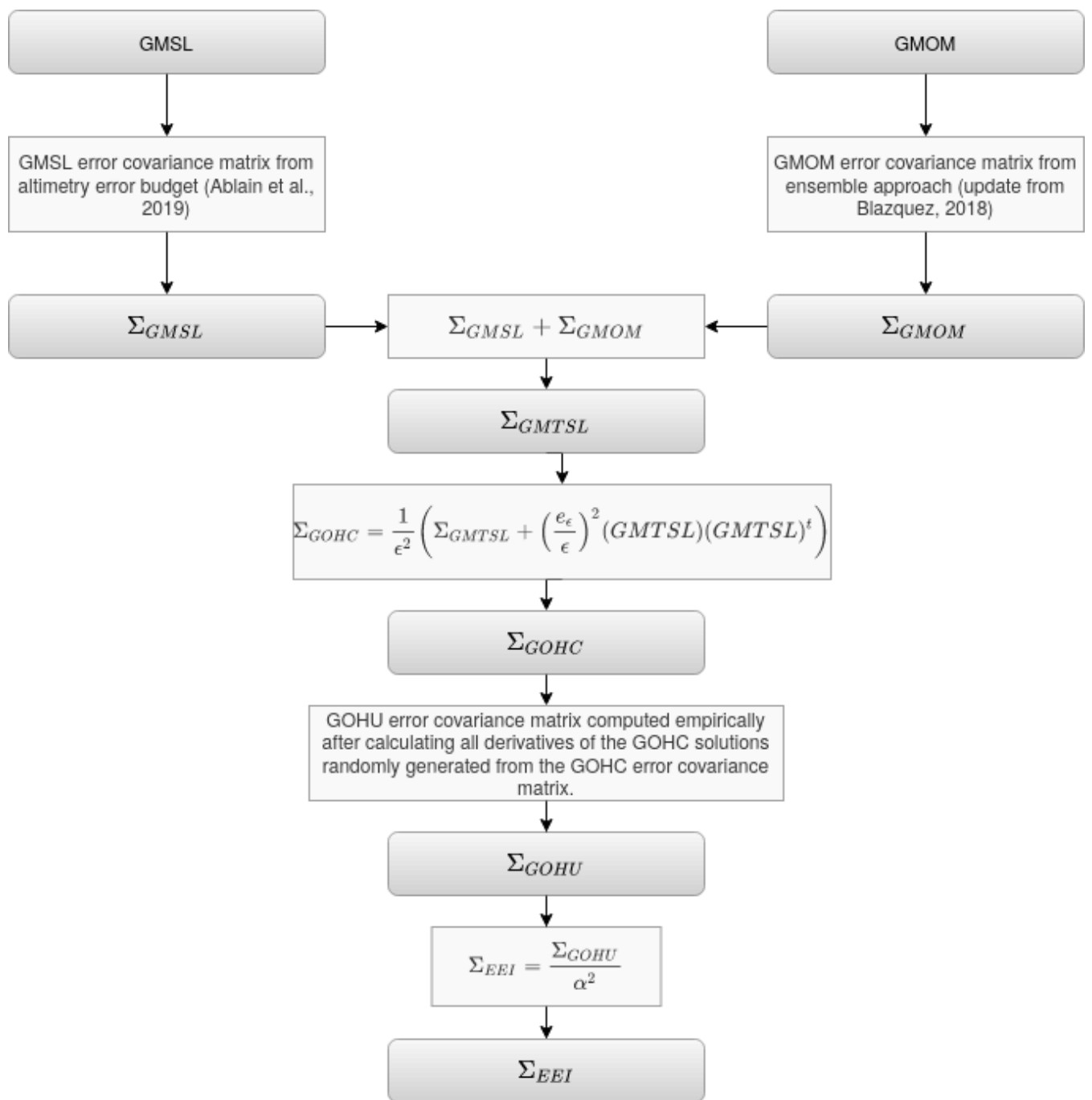

**Figure 2: Propagation of errors from the global mean sea level (GMSL) change and global mean ocean mass (GMOM) change time series until the global ocean heat content (GOHC) change and Earth energy imbalance (EEI) time series.**


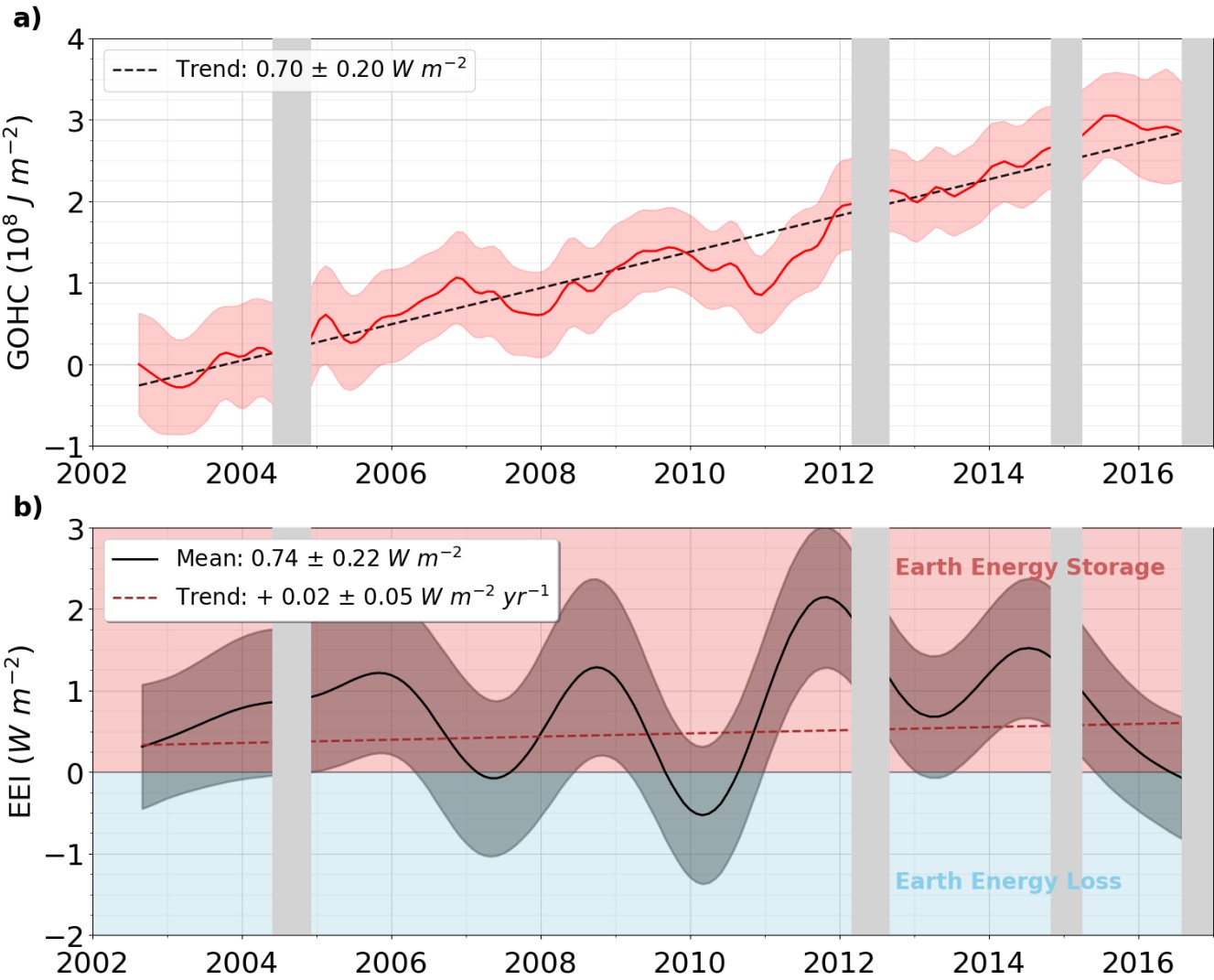

**Figure 3:** Times series of (a) global ocean heat content (GOHC) change and (b) Earth energy imbalance (EEI) from space geodetic approach (MOHeaCAN v2.1) over the August 2002-August 2016 period. Data spatial distribution considered for the GOHC change computation is presented in Fig. 1. The uncertainty envelopes are superimposed (at 1-sigma). Uncertainties on trends and means are reported within a 90 % confidence level (1.65-sigma). The GOHC change curve is shifted along the ordinate axis to start from the origin in 2002. Grey areas correspond to data gaps in the gravimetry product used for the space geodetic GOHC change.


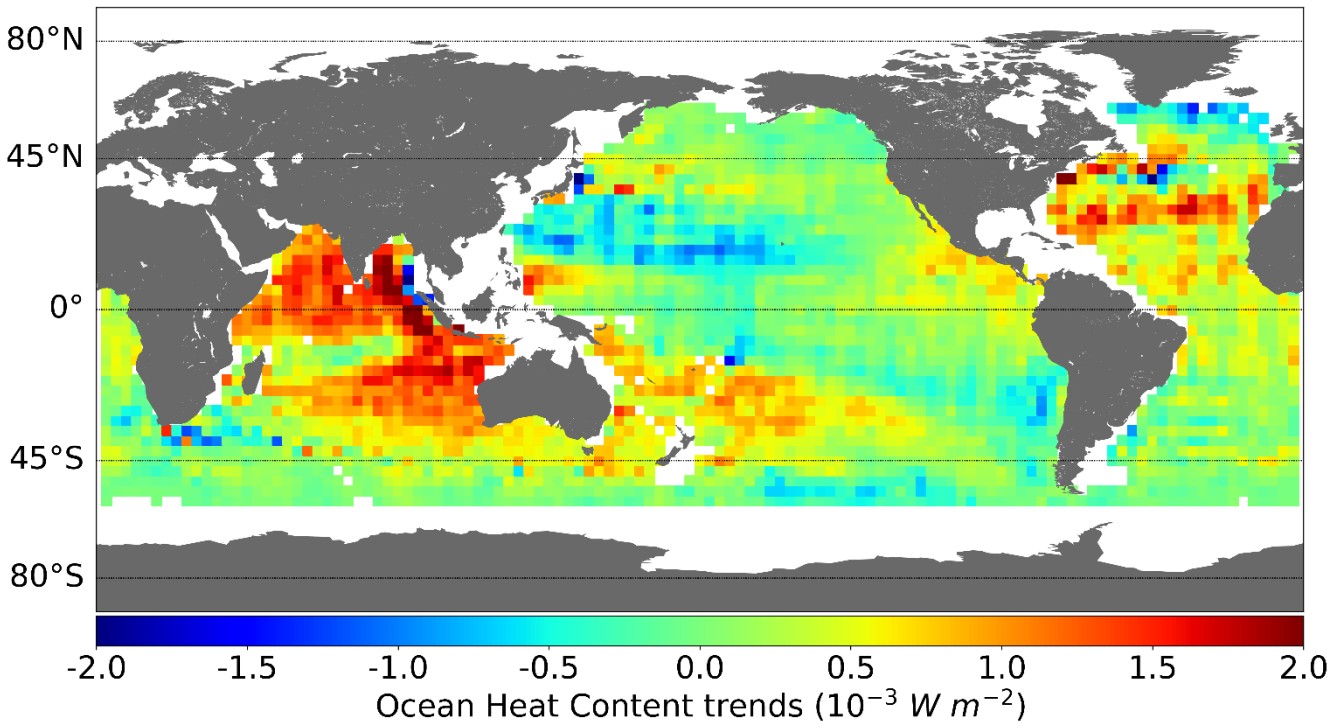

**Figure 4: Map of ocean heat content trends from space geodetic approach (MOHeaCAN v2.1) computed over the August 2002-August 2016 period, 3x3 degree resolution.**

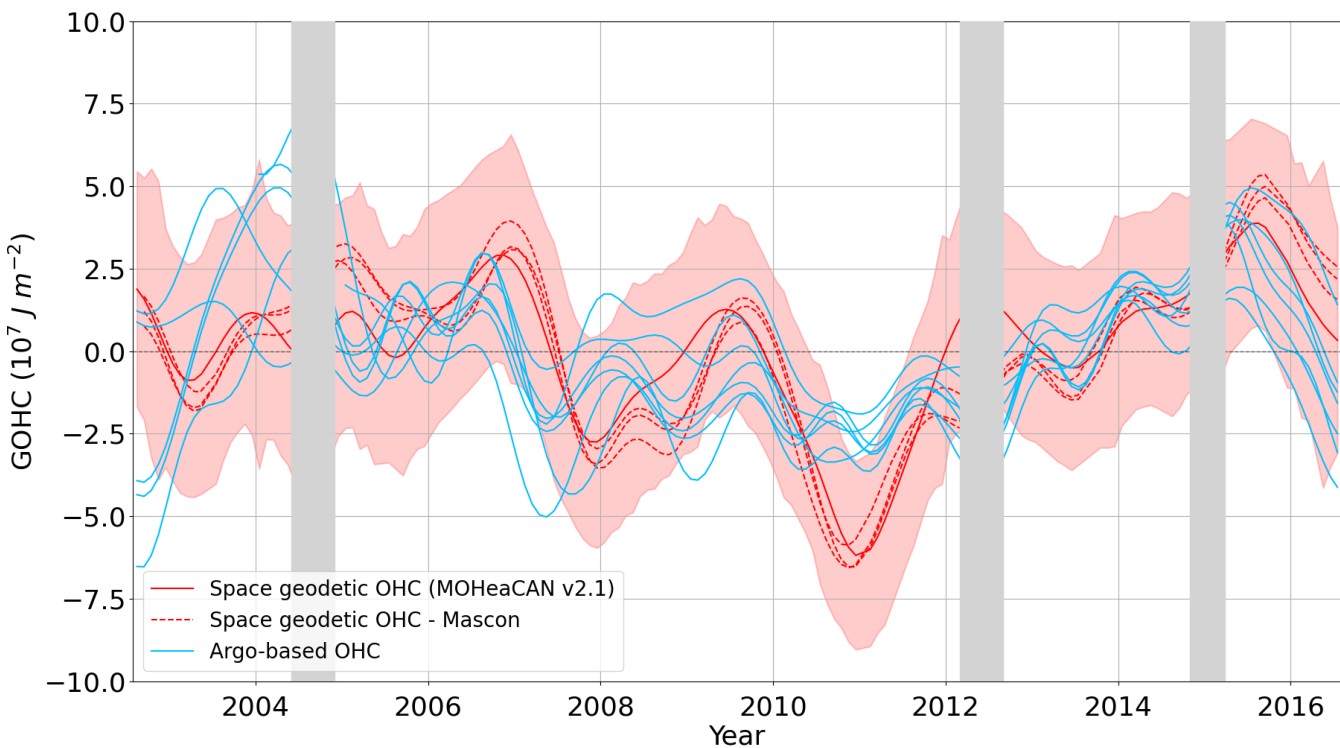


**Figure 5: Interannual variations of global ocean heat content (GOHC) change. A 13-month low-pass filter is applied after removing periodic signals (annual and semi-annual) and trend. Red lines correspond to space geodetic estimates where estimates based on mascon ocean mass are represented in dash lines and MOHeaCAN v2.1 is represented by mean value (solid red) and the uncertainty at 1-sigma (shaded areas). Blue lines correspond to the Argo-based estimates from 2005. Grey areas correspond to the data gaps in**
**the gravimetry product used for the space geodetic GOHC change.**

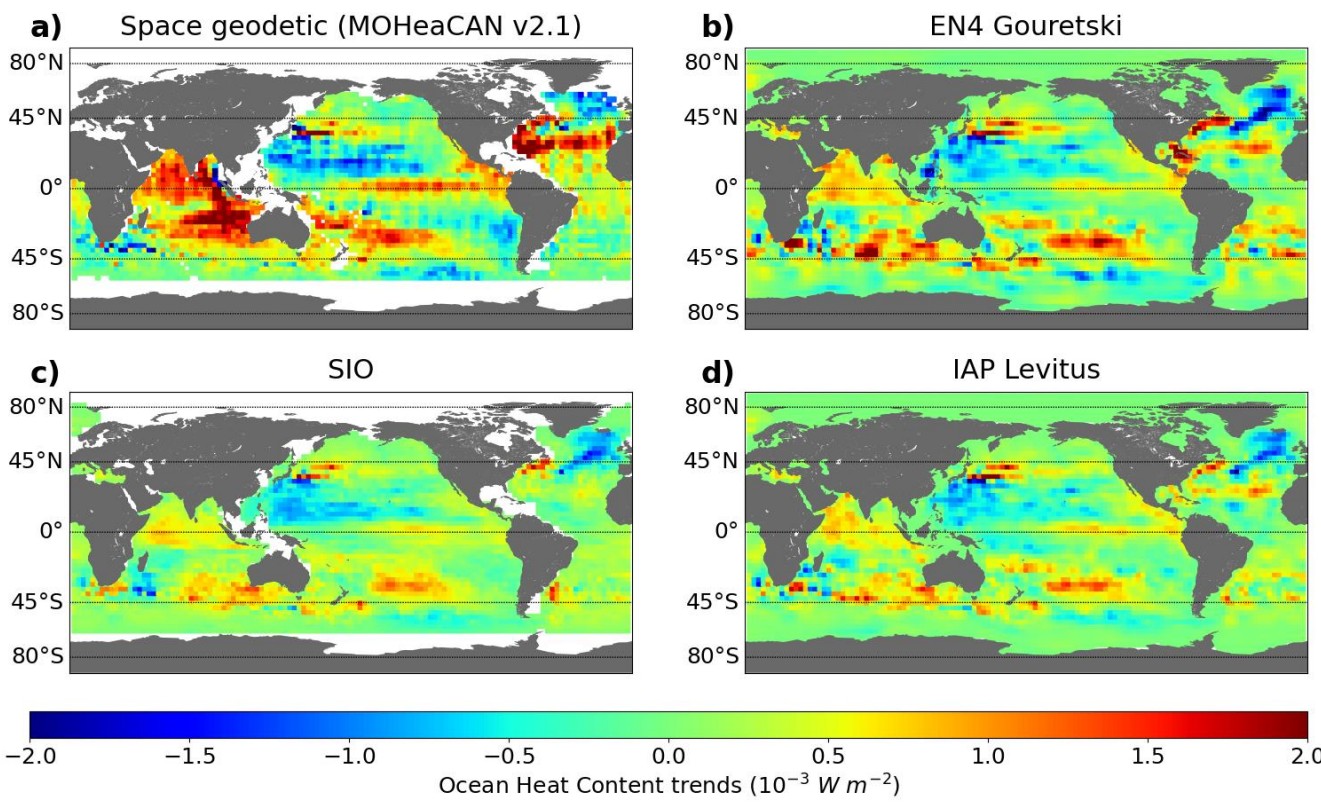

**Figure 6: Maps of ocean heat content trends from space geodetic approach for the period from January 2005 to December 2015 and at 3x3 degree resolution.**

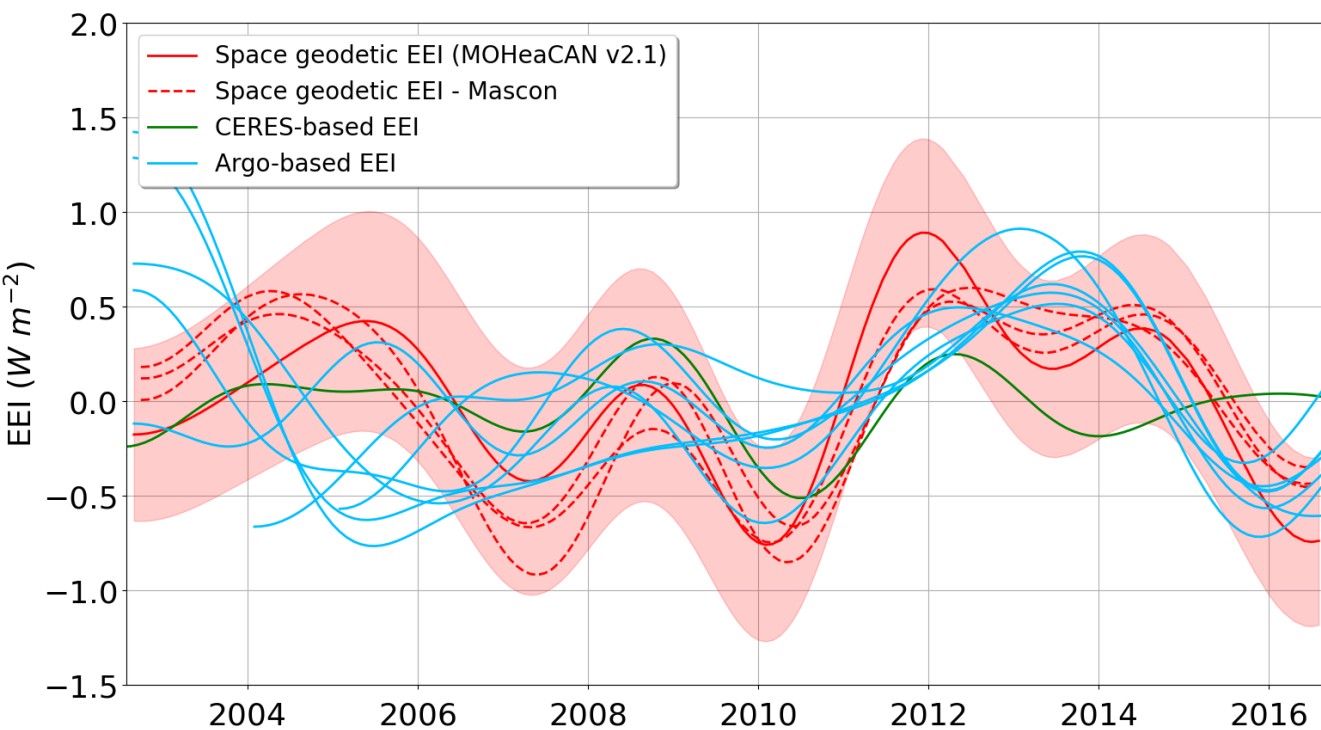

**Figure 7: Interannual variations of Earth energy imbalance (EEI) time series. Mean and trend values have been removed for each time series and a filter has been applied to remove signals lower than 3 years. Red lines correspond to space geodetic estimates where estimates based on mascon ocean mass are represented in dash lines and MOHeaCAN v2.1 is represented by mean value (solid red) and the uncertainty at 1-sigma (shaded areas). Green line corresponds to CERES-based estimates and blue lines to the Argo-based estimates.**

