# Peer review of "Monitoring the ocean heat content change and the Earth energy imbalance from space altimetry and space gravimetry"

_Earth System Science Data, 2021_

## Author Comment (AC1)

**RC#1**

The presented dataset and article are highly relevant for in-depth research on Earth's energy imbalance (EEI), its uncertainty, and causes for (internal) variability. The authors derive ocean heat storage from geodetic satellite observations, namely altimetry and gravimetric observations that lend themselves to derive the ocean's thermal expansion as their residual – at least in the global mean. While this approach has been successful in previous studies to constrain EEI - which by definition is a global metric - the authors go so far as to also present a regional ocean heat storage dataset based on the geodetic data. The gridded heat content/storage data is flawed as the authors admit, but may inspire the community to help improve and produce higher quality products of similar sort.

While the paper is important, well written and places rigor on uncertainty quantification, I have identified a few major weaknesses that I would like to see addressed before publication. Editorial comments are at the end of this review.

- **Expansion efficiency of heat:**
  1. I agree that gridded Argo data are useful to derive the EEH, assuming that the 0-2000m and full column efficiencies are very similar, at least in the global mean. However, by using Argo data in the form of EEH to derive GOHC from the geodetic space observations, your geodetic GOHC estimate is not fully independent from Argo data. How does that affect the validation results against Argo GOHC and EEI? On line 444 and 474 you state the independent nature of the datasets, but this is not truly the case – if I understand correctly, an EEH derived from ocean observations will always be needed to translate satellite-based steric changes to GOHC.

EEH is indeed retrieved from in situ observations. So, strictly speaking, the geodetic GOHC estimate is not independent from Argo. However the dependence is negligible.

In the geodetic approach, we use Argo to estimate the global EEH and we find that Argo solutions provide mostly the same global EEH of 0.145 m/YJ with a range of ±0.7%. This very small range means that the EEH is mostly insensitive to different Argo solutions. This is because the global EEH is a thermodynamic variable which depends, at global scale, only on the warming pattern (i.e. the spatial distribution of absolute T - climatology plus anomaly -) and thus it changes only when the spatial distribution of absolute T changes significantly (See Russel et al. 2000 in Kuhlbrodt and Gregory 2012). Since all Argo products agree mostly on the absolute T, they yield mostly the same EEH estimate and the geodetic approach is mostly insensitive to Argo products (less than 0.7%).

Having said that, we acknowledge that we can not state that the geodetic approach is independent from Argo. We remove this statement from the manuscript and we explain clearly at the end of section 2 that the in situ data is only used to derive a static value and here is why the approach is named "space geodetic".

  2. Your mean EEH value amounts to 0.145 m/YJ (or 0.43 Wm$^{-2}$/mmyr$^{-1}$) and is therefore larger than the Levitus (0.12 m/YJ or 0.52 Wm$^{-2}$/mmyr$^{-1}$) estimate for the 0-2000m ocean column. What is the reason for this? Dealing with such small numbers, I find this discrepancy rather large. If we assume a steric sea level change of 1.6 mm yr$^{-1}$, your EEH yields an ocean heat storage of 0.69 Wm$^{-2}$; but using the Levitus number, the heat storage becomes 0.83 Wm$^{-2}$. I find this difference quite dramatic.

Furthermore, the tiny EEH uncertainty derived here does not acknowledge that a smaller EEH of 0.12 m/YJ might be possible.

3. How is the EEH uncertainty derived? With +- 0.001 m/YJ it is one order of magnitude smaller than the uncertainties by Church et al. and Levitus et al. Why is that?

For the calculation of EEH at global scale, monthly 3D in situ temperature and salinity fields from various 11 Argo solutions were used to compute the ratio between GMTSL change and GOHC change. These monthly ratios are averaged over time, then averaged together to provide a global EEH estimate of $0.145 \pm 0.001$ m YJ$-1$ representative of the 0–2000 m ocean column for the period 2005-2015, excluding marginal seas and areas located above 66° N and 66° S. This regional extent corresponds to the spatial extent that is regularly sampled by the in situ Argo network. The global EEH estimated here is in good agreement with previous estimates of $0.12 \pm 0.01$ m YJ$-1$ (equivalent to 0.52 W m$-2$/mm yr$-1$) representative of the 0–2000 m ocean column over 1955–2010 from in situ observations (Levitus et al., 2012) and $0.15 \pm 0.03$ m YJ$-1$ for the full ocean depth over 1972–2008 (Church et al. 2011). Its uncertainty is however much smaller than in previous studies because our EEH computation is based on the Argo network that has a precise estimate of ocean temperature and salinity down to 2000 m depth and our estimate relies only on effective measurements that were processed homogeneously (eg. interpolated data are excluded, the same horizontal and vertical mask is used). Previous studies from Levitus et al. (2012) and Church et al. (2011) used an ensemble of temperature and salinity products that covered the whole ocean over the past decades with in-filled data where measurements are lacking. The differences in the in-filled data explain the large uncertainty in Levitus et al. (2012) and Church et al. (2011). Here we restricted the study to the region and the time span covered by Argo. We expect estimates of EEH to be very precise when the calculation is restricted over the sampled region because EEH accuracy depends only on T,S measurement accuracy and the TEOS10 equation accuracy (and both are very accurate at levels below 0.1%).

Note that our accurate estimate of EEH does not prevent it to be biased by systematic effects not accounted for. In particular the systematic effect of the sampling of Argo which is not fully global (measurements are sparser above 66° latitude and below 2000m depth). Because of this effect our estimate of the global EEH is likely biased by a few percent. It is likely biased high because the bottom layer, below 2000m depth, is less salty than upper layers which would result in a slightly lower global EEH estimate if it was accounted for in the computation. We dedicated a whole paragraph in the new version of the manuscript to explain this in detail. In particular we identify clearly that our estimate of EEH is precise but potentially biased high (see section 3.3).

Even if the temporal variability in EEH is small as the authors state, I would expect a certain degree of variability, which might impact (increase?) the EEH uncertainty? Calculating the EEH on a monthly basis, I would expect a lot of noise, it might be helpful to see a figure of the EEH timeseries as part of the appendix, and to use this timeseries to derive the uncertainty (in case that is not what the authors already do).

EEH can not be calculated on a monthly basis because on a monthly basis, it occurs that the GOHC change over the entire ocean is null while the global thermosteric sea level change is not. A typical example is when the heat uptake of the ocean above the thermocline is compensated by an equivalent heat loss below the thermocline (induced by a change in the thermohaline circulation for example). In such a case the total heat uptake of the entire ocean is by definition zero but the thermosteric sea level change is >0. This is because the expansion of the sea water above the thermocline (which occurs in warmer water) exceeds the contraction below the thermocline (which occurs in colder water). In such a situation the expansion efficiency of heat is not defined and cannot be calculated (since ΔGOHC=0) (note that this is probably the reason why you get a very noisy EEH estimate if you tried a computation on monthly time scales).

A way around this issue is to consider time scales longer than 1 month. At interannual time scales >3-5 years the GOHC increase is such that ΔGOHC>0 and global EEH is always defined. Another way around this issue is to consider the integrated expansion efficiency of heat (IEEH) as we did for the regional OHU (see the new version of the manuscript section 3.3) and derive from it the time variations in EEH. Both approaches, when used with an equivalent level of smoothing, lead to the same variations in EEH. We find interannual to decadal variations in EEH does not exceed 2e-28 m/J (see figure S1 below). This is about 3 orders of magnitude smaller than the mean EEH and thus it is totally negligible in the uncertainty estimate.

[Figure]

**Figure S1: EEH in m/J computed over time scales>3years**

4. Figure 1, spatial distribution of EEH: Something might be wrong with this figure. EEH increases with temperature and salinity. This means EEH must be high in the tropics/sub tropics and low near the poles. At the moment, the map depicts a pattern that is opposite to expectations.

That is true, this figure does not depict space variations of EEH but space variations of the Integrated Expansion efficiency of heat.

At regional scale, it occurs for a water column that the OHC change over the entire column is null while the thermosteric sea level change is not. A typical example is when the heat uptake of a water column above the thermocline is compensated by an equivalent heat loss below the thermocline. In such a case the total heat uptake of the entire water column is by definition zero but the thermosteric sea level change is >0. This is because the expansion of the sea water above the thermocline (which occurs in warmer water) exceeds the contraction below the thermocline (which occurs in colder water). In such a situation the EEH is not defined and cannot be calculated. A way around this issue is to consider ocean heat content (OHC) (rather than ocean content changes ΔOHC) and thermosteric sea level (TSL) (rather than thermosteric sea level changes ΔTSL) and to define an integrated expansion efficiency of heat (IEEH) $\mathbf{E}$ as follows:

$\mathbf{E}$=TSL/OHC. (1)

The IEEH is in m.J-1 as the EEH. At regional scale, the IEEH is always calculable because the ocean heat content is never null. Thus, the IEEH allows to derive estimates of regional ocean heat uptake (OHU) from estimates of the regional thermosteric sea level (TSL) with the following equation:

$$OHU=dOHC/dt=d(\mathbf{E}.TSL)/dt. \qquad (2)$$

In our work we use the classical definition of EEH to derive estimates of the GOHU and the EEI. However we use equations 1 and 2 to derive estimates of the regional ocean heat uptake OHU. We verify the consistency of the global and regional estimates of the ocean heat uptake by comparing the global sum of OHU with GOHU.

This detail was not clear in the precedent version of the manuscript. Now it is clarified (See section 2 of the new manuscript).

As you noted rightly, Figure 1 does not represent the spatial variations in EEH. It represents spatial variations in IEEH. The IEEH is different from the EEH. IEEH expresses the change in thermosteric sea level due to a change in ocean heat content. It represents the ratio of the thermosteric sea level over the heat content. As such it allows estimating OHC from thermosteric sea level (following Eq. (6) of the manuscript). The IEEH can be calculated from known ocean variables (IOC et al. 2010) as the specific volume (m3/kg) divided by the specific enthalpy (J/kg). The IEEH is dependent on temperature, salinity and pressure, it increases with temperature and pressure and decreases with salinity (see Figure S2). Note that, because IEEH decreases with salinity while EEH increases with salinity, when integrated over the entire water column, the spatial variations of the IEEH are expected to be different from the spatial variations in EEH. IEEH increases with latitude as the total salinity content of water columns decreases with latitude. This is now clarified in the new version of the manuscript: see section 3.3.

[Figure]

**Figure S2: Integrated expansion efficiency of heat (IEEH) dependence on in situ temperature and absolute salinity at 1 atm in mm/J-1.**

- **Calculations & geophysical corrections:**
    1. I would appreciate a table that summarizes the values for geocentric GMSL, GMSL (after correction), GOML, EEI etc. Without the table, it has been difficult to follow your calculations. For example, the value for the GMTSL is not provided in the paper.

→ Such a table was added in the appendix. It is referred to in section 5.1 and 6 and presents all the trends (and mean for the EEI variable) for both periods (study and comparison).

2. I backed out the GMTSL value using the authors' EEH= 0.43 $Wm^{-2}/mmyr^{-1}$ and GHOC (2002-2016) = 0.7 $Wm^{-2}$. The GMTSL should be 1.63 $mmyr^{-1}$. With this and the GMOM trend of 1.83 $mmyr^{-2}$, I arrive at an GMSL trend of 3.46 $mmyr^{-1}$. For the same period, the authors report a GMSL trend of 3.57 $mmyr^{-2}$. Please rectify the 0.1 $mmyr^{-1}$

There are no errors in the paper on the different trends presented. However, there is a lack of information on the products used to calculate the different GMSL trends that led to this comment:

In section 3.1 we presented the values of the GMSL trends from the AVISO GMSL indicator from the L2P along-track products (MSL_Serie_MERGED_Global_AVISO_GIA_Adjust_Filter2m downloaded on the AVISO website). Over the study period this leads to a trend of 3.57 mm/yr (once the GRD correction is considered).

Using the C3S gridded products as input dataset for the MOHeaCAN product, the trend obtained (also adjusted for GIA and GRD with the good sign) is 3.49 mm/yr: it is in agreement with the reviewer calculation accounting for the very slight approximation on the EEH global value.

The difference of 0.08 mm/yr in GMSL trends between along-track L2P products and C3S gridded products does not come from an error on the GRD signal, but from the difference between along-track and gridded products. This difference remains low and within the GMSL trend uncertainty (0.3 mm/yr).

→ In order to limit confusion, we specified in section 3.1 that GMSL trends are computed from the AVISO time series. Readers can now find the C3S GMSL trend in the table given in the appendix (question #1). There is no need to rectify the 0.1 mm/yr as requested by the reviewer and the table should clarify the consistency of the trends.

3. The authors are not reporting the geocentric GMSL change. Adding to their reported (relative) GMSL trend (3.57 $mmyr^{-2}$) the GIA and ocean bottom deformation - which should sum up to -0.38 $mmyr^{-1}$ - the geocentric GMSL must be near 3.19 $mmyr^{-1}$. This seems very low. I would expect a trend of ~3.5 $mmyr^{-1}$ according to the AVISO data. I suspect the authors might have confused the corrections. The sign of the correction for ocean floor deformation should be negative (-0.1 $mmyr^{-1}$) so that: GMSL = geocentric GMSL– GIA – GRD = geocentric GMSL + 0.38 $mmyr^{-1}$

The error detected by the reviewer concerns a misprint error on the sign of correction of the GRD but the GMSL calculated in the paper is calculated with the good GRD and GIA sign. So there is no error on the GMSL trend presented in the paper. For more explanation see our answer for comment #2 that may have misled the reviewer.

→ We have corrected the misprint on the GRD correction sign.

4. By the way, GMTSL is not spelled out, but I believe it refers to Global Mean Total Steric sea Level?

GMTSL refers to Global mean thermosteric sea level. This acronym is introduced in section 2 (as for the GMOM and GMSL acronyms).

- **EEI uncertainties:**
  1. I appreciate the authors thinking about the impact high frequencies in GOHC variability have on EEI derivation. However, the authors do not justify the low-pass filter cut-off period of 3 years sufficiently. Lines 100-103 would need a reference and maybe even a demonstration of what the filter does to the GOHC time series. Furthermore, I wonder what slightly different filters or cut-off periods may look like? Did the authors do an assessment of different filters? How does EEI look like if no filters are applied? I understand such an assessment might be beyond the scope of this article, but I think this matter is important and needs to be addressed, at least briefly.

As explained in the paper, the GOHC change is filtered out to remove the signals related to the intrinsic climate variability. These signals are mostly generated in the mixed layer above the pycnocline in response to exchanges with the atmosphere caused by climate modes of variability (such as ENSO). The 3-year cut-off period chosen in this study comes from an external study based on climate models: Palmer and McNeall (2014).

We have also calculated the EEI without filtering out the GOHC time series, or changing the cut-off period to 1 year and 2 years. In these cases, we obtained EEI time series with variations significantly higher than with the 3-year cut-off period (no filter: -24 to 25 W/m²; 1-year cut-off period: -2 to 4 W/m² ; 2-year cut-off period: -2 to 3 W/m²). However these variations for short time scales (< 2-3 years) do not correspond to any response to global warming and therefore must be removed to infer variations in the EEI. In our opinion, there is no interest to show them in this paper.

→ We have clarified the text by adding a reference to Palmer and McNeall (2014) who explain these time scales, justifying the 3-year cut-off period applied in this study (reference added in sections 2 and 4.1).

  2. On line 248, the authors state the high frequency (to be filtered out) is only visible in the altimetry and not the gravimetry data. Can this be shown in a figure in the appendix? Would it be better to only filter GMSL before combining with GMOM?

Indeed, the filtering of altimetry and gravity measurements separately before combining them could be of interest to improve in particular the spatial resolution of the OHC grids, but also to try to filter more efficiently the internal ocean variability. At this stage of the study, we have considered this work as a perspective.

→ We have added a few lines in the conclusions section to explain that further studies are necessary to identify which filters (spatial and temporal) should be applied to get better regional OHC change and EEI estimates.

  3. By applying the smoothing filter to GOHC before deriving EEI, the authors had to empirically estimate EEI uncertainty. What would the EEI uncertainty be if no filter was applied and the errors could be propagated as for GOHC? I expect the uncertainty is sensitive to the filter applied, which means the error due to the filtering needs to be included? Perhaps, this could be based on a sensitivity analysis (using different filters/cut-offs). For this article, a discussion about the impact of filtering GOHC on the EEI uncertainty estimate would be appreciated.

First, our calculation of the uncertainty of the EEI does not depend on the filter applied, and in general on the method applied on the spatial geodetic data. We invite the reviewer to read our detailed arguments in our response to comment #4.

Although the results are not presented in the paper, we have already performed such a sensitivity study. Reducing the GOHC low-pass filter cutoff frequency ($\lambda$) has this impact on the mean EEI uncertainty ($\sigma$) in a 68% confidence interval (1-$\sigma$):

- No filter:     $\sigma$= 0.22 W m$^{-2}$
- $\lambda$= 1 year:     $\sigma$= 0.18 W m$^{-2}$
- $\lambda$ = 2 years:  $\sigma$= 0.15 W m$^{-2}$
- $\lambda$ = 3 years:  $\sigma$=  0.14 W m$^{-2}$

The mean EEI uncertainty estimates increase slightly with no filtering or a lower cut-off period. This is due to the high frequency errors (< 3 years) in the spatial geodetic measurements, which are also filtered out to construct the variance-covariance matrix of the EEI ($\Sigma$): see section 4.2 of the paper for more detail on the method applied.

However, we chose not to present these results because, as explained in the previous comment, we filtered the GOHC time series with a cutoff period of 3 years in order to remove signals related to internal ocean variability in the EEI time series. Furthermore, the EEI uncertainty estimates are significantly underestimated for $\lambda$ < 3 years because the internal ocean variability signal is not described as a source of errors in the error variance-covariance matrix of GOHC and EEI time series. For $\lambda$ = 3 years, this simplification is acceptable, because the ocean internal variability has been removed. Thus, at this stage of our study, there is very little point in providing the filtered EEI calculation with a cutoff period of less than 3 years and such an uncertainty sensitivity calculation.

> 4. Compared to Meyssignac et al. (2019), the EEI uncertainty seems to have almost halved. I do not agree that this is solely due to the decreased EEH uncertainty - and even if, I suspect the EEH uncertainty is underestimated I think the role of the low-pass filter needs to be unveiled. I recommend to discuss and summarize potential reasons as to why the uncertainty range has decreased.

The EEI mean uncertainty [5%-95%] has not exactly halved in this paper compared to Meyssignac et al. (2019), as over a similar period:

- Meyssignac et al. (2019): 0.57 +-0.29 W/m² (over 2002-2016)

- Marti et al. (2019) : 0.70+-0.20 W/m² (over August 2002- August 2016)

First of all the uncertainty claimed in Meyssignac et al. (2019) is based on preliminary uncertainty estimates under some hypothesis and not calculated with a formal error propagation as it is proposed in this study. One of the strong added value in this paper, is the formalism proposed to propagate for the first time the space geodetic measurements errors until the GOHC and EEI.

This being said, the reduction of the uncertainties linked to the new EEH calculation has a significant impact since we reduce the uncertainties on the GOHC trend from 0.27 to 0.20 W/m² (08-2002-06/2002) by considering an uncertainty of 0.03 to 0.001 m/YJ on the EEH.

In addition, our uncertainty estimates do not account for the sensitivity of the method to the averaging and collocation of spatial geodetic data, or to the filtering of the GOHC time series. Although the calculation of OHC and EEI is sensitive to the definition of the method, we focused at this stage on the uncertainty only related to the propagation of the measurement errors from space geodetic observations to OHC and EEI times series

assuming that the method or the space data sampling (in space and in time) is not a source of error. The same approach was done for the sea-level rise uncertainty calculation at global and regional scales by Ablain et al. (2019) and Prandi et al. (2021) where only altimetry measurement errors are considered. However, it would be relevant in a second step to measure the sensitivity of the method (e.g. averaging, filtering), but also the fact that the EEH does not represent the full-depth ocean (deep-ocean and coastal areas), or the fact that the space geodetic measurements do not measure the very high latitudes.

→ Those last elements are already mentioned in the conclusion section as the perspectives.

- **Regional OHC:**
    1. As the authors state correctly, the regional OHC estimates from geodetic data require a correction for regional halosteric effects. My question is then, instead of comparing OHC between geodetic and in-situ data, why not compare the total steric changes instead? I understand there is an issue with salinity drift in some or most of the in-situ datasets. But there are some, e.g., the SIO dataset, which do not show a global drift in halosteric sea level change. I think using steric changes would be a clearer and less convoluted comparison between the geodetic and in-situ retrievals.

We do agree that for the purpose of validating the in-situ data and the space geodetic data, it would be more direct to compare the regional steric sea level. But the objective of this paper is to give a first regional estimate of the OHC changes, although these have been calculated with simplifying assumptions, with in particular the non-accounting of the spatial salinity changes. Thus, the primary objective is not so much to verify the consistency of the different data used, but to show how far we still have to go to improve these regional OHC estimates and to understand the differences between the different data sets. This is what we have indicated in the perspectives of this study.

→ We modified the manuscript, used "comparison" instead of "validation"

    2. Line 253: I do not understand how the salinity effect is accounted for by the EEH map over 2005-2015. The EEH is defined as the ratio of thermosteric (sea level rise due to thermal expansion) over OHC changes. You would need thermosteric sea level change to back out OHC using these EEH estimates, but the geodetic data provide you with total steric changes. No matter the time period, you would need information on the regional halosteric changes.

The integrated expansion efficiency of heat is computed with the data of 11 Argo products over 2005-2015 (we start in 2005 because the coverage of Argo is global in 2005. We end in 2015 because salinity issues in the Argo products start at the end of 2015 see for example Ponte et al. 2021). We use the mean salinity field of the 11 Argo products over 2005-2015 to estimate the IEEH (instead of the reference salinity at 35psu). So the IEEH is computed taking into account the density of sea water at the level of the 2005-2015 mean salinity. It means that the density effect (i.e. the halosteric effect) of the mean salinity over 2005-2015 is accounted for in IEEH. However any halosteric effect induced by a change of salinity wrt to the mean over 2005-2015 is not accounted for. So the map, that is computed as a mean over 2005-2015 accounts for salinity but any other map, computed over a different period, would miss a small halosteric contribution from salinity anomaly wrt to the mean salinity over 2005-2015.

→ The explanations were reformulated and completed.

3. I do not believe the regional/gridded OHC as it stands is useful for the community to study OHC variability. This article and the public datasets require a disclaimer on the use of the gridded OHC derived from space geodesy. Or the authors could provide the total steric changes instead or in addition to the OHC map.

We agree with the reviewer that the regional OHC changes provided in this study need more work to be fully used by the scientific community. But as explained earlier, our goal is to provide a first regional estimate of OHC geodetic space changes and to show how far we have to go to improve these regional OHC estimates and to understand the differences between different datasets. Providing the steric sea level changes is also very relevant of course, but this has already been done by other authors and does not fit our purpose here. We now warn the reader twice in the new version of the manuscript on the use of the OHC grids .

In a future work we intend to evaluate the error in OHC that is induced by neglecting the salinity variations at regional scale. We expect the effect to be small (less than 10% of the signal locally) as the salinity changes are quite small in % of the total salinity signal and their impact on the total sea level is even smaller in terms of % of the total sea level signal. The effect of salinity should be visible only in regions of large salinity variability as over the shelf of the Arctic ocean.

4. Besides, I doubt the EEH map (Figure 1) is correct, which affects the gridded OHC. By providing the steric changes instead (or separately with the EEH map), users could use their own EEH maps and assumptions to derive OHC.

That is true, this figure does not depict space variations of EEH but space variations of the Integrated Expansion efficiency of heat.

At regional scale, it occurs for a water column that the OHC change over the entire column is null while the thermosteric sea level change is not. A typical example is when the heat uptake of a water column above the thermocline is compensated by an equivalent heat loss below the thermocline. In such a case the total heat uptake of the entire water column is by definition zero but the thermosteric sea level change is >0. This is because the expansion of the sea water above the thermocline (which occurs in warmer water) exceeds the contraction below the thermocline (which occurs in colder water). In such a situation the EEH is not defined and cannot be calculated. A way around this issue is to consider ocean heat content (OHC) (rather than ocean content changes ΔOHC) and thermosteric sea level (TSL) (rather than thermosteric sea level changes ΔTSL) and to define an integrated expansion efficiency of heat (IEEH) $\mathbf{E}$ as follows:

$$\mathbf{E}=TSL/OHC. \quad (1)$$

The IEEH is in m.J-1 as the EEH. At regional scale, the IEEH is always calculable because the ocean heat content is never null. Thus, the IEEH allows to derive estimates of regional ocean heat uptake (OHU) from estimates of the regional thermosteric sea level (TSL) with the following equation:

$$OHU=dOHC/dt=d(\mathbf{E}.TSL)/dt. \qquad (2)$$

In our work we use the classical definition of EEH to derive estimates of the GOHU and the EEI. However we use equations 1 and 2 to derive estimates of the regional ocean heat uptake OHU. We verify the consistency of the global and regional estimates of the ocean heat uptake by comparing the global sum of OHU with GOHU.

This detail was not clear in the precedent version of the manuscript. Now it is clarified (See section 2 of the new manuscript).

As you noted rightly, Figure 1 does not represent the spatial variations in EEH. It represents spatial variations in IEEH. The IEEH is different from the EEH. IEEH expresses the change in thermosteric sea level due to a change in ocean heat content. It represents the ratio of the thermosteric sea level over the heat content. As such it allows estimating OHC from thermosteric sea level (following Eq. (6) of the manuscript). The IEEH can be calculated from known ocean variables (IOC et al. 2010) as the specific volume (m3/kg) divided by the specific enthalpy (J/kg). The IEEH is dependent on temperature, salinity and pressure, it increases with temperature and pressure and decreases with salinity (see Figure S2). Note that, because IEEH decreases with salinity while EEH increases with salinity, when integrated over the entire water column, the spatial variations of the IEEH are expected to be different from the spatial variations in EEH. IEEH increases with latitude as the total salinity content of water columns decreases with latitude. This is now clarified in the new version of the manuscript: see section 3.3.

That said, users have the option of using their own IEEH map to derive the OHC because the product delivered with this paper contains the steric sea level grids, as well as our IEEH grid.

5. Line 447: I do not remember seeing a quantification of the high correlation of the OHC map with climate mode fingerprints in models or with the Argo data. Has this been assessed? If not, I would weaken the statement by saying the patterns look/appear similar qualitatively. At the very least, this article should show the Argo-based map of OHC change to be able to qualitatively compare the geodetic approach against, which it does not in its present form.

Figure 6 presents various regional OHC trends maps, in particular the results obtained with the space geodetic approach and those derived from in situ data. It provides a first qualitative assessment of the space geodetic based regional OHC. Given the assumptions made when computing the regional OHC changes, a quantitative comparison against in situ based results, with spatial correlation map for instance, appears premature. With regards to climate mode fingerprints, we rely on Pfeffer et al., 2018 and indeed mention only a qualitative comparison.

→ We reformulated the text in line 447 in agreement with the reviewer's suggestion.

**Editorial and Minor comments:**

- line 16: "… estimating the ocean heat content (OHC) change provides an accurate proxy of the EEI." corrected
- line 29: "It is challenging to estimate the EEI from TOA radiation fluxes since it is two orders of magnitude smaller…" corrected
- line 37: delete "counted" corrected
- line 43: "The direct measurement approach relies…" corrected
- line 48: "The ocean net flux approach assesses the radiative …" corrected
- line 54: "except for the polar regions" corrected
- line 56-59: This sentence is long and convoluted, please edit. This has been rephrased.
- line 68: "Reducing uncertainty as much as possible." Can you explain/provide examples of how to reduce uncertainties? I'm a bit taken aback by this statement. I hope the message is not to make the uncertainty look smaller than it is by favoring certain approach over others? This has been rephrased. A full uncertainty propagation scheme is likely to help reduce the uncertainties, otherwise, conservative assumptions are often made.
- line 68: What do the authors mean by "extending the spatial and temporal coverage of OHC change? What exactly is being extended and how?
- line 71: change not changes corrected
- line 112: referred to as corrected

- line 117: "for" not "from" corrected
- line 121: - 1 mmyr[-1] sign corrected (- 0.1 mm/yr)
- line 160: How do you make the post-processing choices? Do you account for the probability of the choices? How?

  The spherical harmonics solutions ensemble is an update of the ensemble described in Blazquez et al., 2018. Depending on the post-processing parameters, the choice of solution may be larger or just one solution. We decided to include all published solutions. For some parameters as GIA models we only use global solutions that conserve the total amount of water. For other parameters such as Earth's oblateness we only include 2 solutions as the spread among the solutions is one order of magnitude smaller than the rest of post-processing parameters. As there is no scientific evidence to prefer one solution over another, we assume each solution as equally probable. Then we assume that the uncertainty in GRACE-based ocean mass estimate should be represented by the uncertainty in these post-processing parameters.

- line 163: What are these state-of-the-art estimates? References?
- GRACE ocean mass estimates based on Blazquez et al., 2018 were compared to several other solutions in terms of trends, see Figure S3 below.

[Figure]

**S3: ocean mass change trends derived from various recent studies.**

- line 194: Please indicate here that the regional OHC derived does not include a correction for regional halosteric effects and should be used with caution. → comment added in section 4.1
- Line 242: "taking" not "making the difference" corrected
- Line 244: please state the surface area in numbers corrected
- Line 250: "Monthly steric sea level changes are directly…" delete "grids" corrected
- Line 262: I was under the impression EEH was calculated as the ratio of thermo-steric sea level change and OHC change. How exactly do you incorporate salinity climatologies? The IEEH expresses the change in thermosteric sea level due to a change in ocean heat content. It represents the ratio of the thermosteric sea level over the heat content. As such it allows estimating OHC from thermosteric sea level (following Eq. (6)). The IEEH can be calculated from known ocean variables (IOC et al. 2010) as the specific volume (m3/kg) divided by the specific enthalpy (J/kg). The specific volume depends on salinity and also the specific enthalpy. That is where we incorporate the mean salinity field over 2005-2015
- Line 264: What is meant by "counterpart" – do you mean "downside"? corrected
- Line 274: delete "consisting in" corrected
- Line 276: first step consists of corrected
- Line 291: Please spell out GMTSL GMTSL is defined in section 2.
- Line 316 "the final operation applies the formulation…": corrected
- Line 344: What about comparing the GMOM trend against independent estimates of land ice melt and land water storage change? Independent melt estimates for the Greenland and Antarctic ice sheets as well as glacier do exit in the literature, e.g., the IMBIE project. I expect sea level budget assessments like for example by the WCRP 2018 provide estimates of these melt rates and should be able to indicate closure of the ocean mass budget.

Budget assessment is definitely an interesting approach. However, it would raise discussions that are beyond the scope of this paper. Such an approach is limited by the uncertainties of the models. In particular, large discrepancies are observed between the available models of terrestrial water storage variations. Such comparison would deserve a dedicated study with a thorough estimation of the uncertainties of each component.

- Line 408: "integrated over". Delete "For information," corrected
- Line 409: power plants: corrected
- Line 415: I do not understand how this conclusion can be drawn. Monthly variability has been smoothed out and the monthly variations are not shown. If this is discussed here, a figure would help. And how do you conclude that an uncertainty of 0.8 to 1 is large? Compared to what? CERES? Please clarify. The uncertainty is high with respect to the requirements described in the introduction to enable monitoring the response of EEI to anthropogenic and natural forcing. -> The corresponding sentences in the introduction and in section 6 have been rephrased to improve clarity.
- Line 428: "… important signals that sign at global scale." I do not follow, please edit. -> rephrased
- Line 439: I do not fully agree that this is the first study of its kind. There has been work by Llovel et al. (2014), Dieng et al. (2015), and others making use of geodetic datasets to estimate the oceans' thermal expansion. The main novelty of the present work is the implementation of a robust algorithm to propagate the uncertainties. -> We have rephrased the sentence to make this point clearer.
- Lines 473-478: I would delete the sentence "It enables independent comparisons which is the unique method to robustly check and validate the final EEI estimates". It does not add much. -> The sentence has been removed.

---

## Author Comment (AC2)

**Review #2**

**General comments**

The presented study provides estimates of ocean heat uptake and the Earth's energy imbalance, mainly focusing on the space geodetic retrieval (from radar altimetry and GRACE gravimetry). The topic is timely and important, and the material in the paper provides a good basis for further scientific discussions and explorations. Emphasis is laid on the more exhaustive error analysis compared to previous research of the same topic.

IMHO, the paper still contains areas for improvement before a publication is justified. I'll highlight these below.

recommendation to editor: Major revision

**Main points:**

* How does the choice of averaging region affect the estimates? I realize that the inclination and Argo coverage prohibit high latitude estimates, but the consequence is that the reported errors do not take into account the omission of the high latitude signals. Since the main selling point of this paper is the computation of the errors, it would actually be an added value if the authors could try to quantify this high latitude omission error in a (stochastic, i.e. from models) way. I realize this adds extra work, but I think it would be very fitting in this paper.

Our uncertainty estimates indeed do not account for the sensitivity of the method to the averaging and collocation of spatial geodetic data, or to the filtering of the GOHC time series. We focused at this stage on the uncertainty only related to the propagation of the measurement errors from space geodetic measurements to OHC and EEI times series assuming that the method or the space data sampling (in space and in time) is not a source of error. The same approach was done for the sea-level rise uncertainty calculation at global and regional scales by Ablain et al. (2019) and Prandi et al. (2021) where only altimetry measurement errors are considered. However, it would be relevant in the future to measure the sensitivity of the method (e.g. averaging, filtering), but also the fact that the EEH does not represent the deep ocean or is not given at the very near coast, or the fact that the geodetic space measurements do not measure the very high latitudes. This is what we have proposed in the perspective of this study.

* Related to this, is that the restoring of the ocean/atmosphere products in GRACE and consequently subtracting the estimates from IB corrected altimetry should be described in more detail as it can falsely introduce atmospheric signals in the averages of global ocean mass. Are there still ocean averaged atmospheric components in the GRACE data? Or have these been corrected before comparing with altimetry?

GRACE solutions ensemble is corrected from atmospheric effects using the spatial mean of the GAD product. Therefore, the GRACE data provide an estimate of the ocean mass variations, theoretically free from atmospheric effects. However, the correction applied is only as accurate as the model used. If there were significant atmospheric effects not accounted for by the GAD, they would remain in the estimated ocean mass changes. These sources of error may not be adequately described in our ensemble, which may require attention in future studies. Details are available in the Appendix A - l487.

* Uncertainties of EEH. The method to estimate the expansion efficiency of heat comes from a paper which I don't (yet) have access to, so more clarifications may be needed. For example, it is claimed that the EEH is most sensitive to salinity and at the same time it is claimed that the error bars are reduced because of the Argo data. Argo data is known to have considerable biases and errors in the salinity estimates so I wonder whether the authors could better clarify why they think the errors are now considerable smaller.

For the calculation of EEH at global scale, monthly 3D in situ temperature and salinity fields from various 11 Argo solutions were used to compute the ratio between GMTSL change and GOHC change. These monthly ratios are averaged over time, then averaged together to provide a global EEH estimate of 0.145 ± 0.001 m YJ–1 representative of the 0–2000 m ocean column for the period 2005-2015, excluding marginal seas and areas located above 66° N and 66° S. This regional extent corresponds to the spatial extent that is regularly sampled by the in situ Argo network. The global EEH estimated here is in good agreement with previous estimates of 0.12 ± 0.01 m YJ–1 (equivalent to 0.52 W m–2/mm yr–1) representative of the 0–2000 m ocean column over 1955–2010 from in situ observations (Levitus et al., 2012) and 0.15 ± 0.03 m YJ–1 for the full ocean depth over 1972–2008 (Church et al. 2011). Its uncertainty is however much smaller than in previous studies because our EEH computation is based on the Argo network that has a precise estimate of ocean temperature and salinity down to 2000 m depth and our estimate relies only on effective measurements that were processed homogeneously (eg. interpolated data are excluded, the same horizontal and vertical mask is used). Previous studies from Levitus et al. (2012) and Church et al. (2011) used an ensemble of temperature and salinity products that covered the whole ocean over the past decades with in-filled data where measurements are lacking. The differences in the in-filled data explain the large uncertainty in Levitus et al. (2012) and Church et al. (2011). Here we restricted the study to the region and the time span covered by Argo. We expect estimates of EEH to be very precise when the calculation is restricted over the sampled region because EEH accuracy depends only on T,S measurement accuracy and the TEOS10 equation accuracy (and both are very accurate at levels below 0.1%).

Note that our accurate estimate of EEH does not prevent it to be biased by systematic effect not accounted for. In particular the systematic effect of the sampling of Argo which is not fully global (measurements are sparser above 66° latitude and below 2000m depth). Because of this effect our estimate of the global EEH is likely biased by a few percent. It is likely biased high because the bottom layer, below 2000m depth, is less salty than upper layers which would result in a slightly lower global EEH estimate if it was accounted for in the computation. We dedicated a whole paragraph in the new version of the manuscript to explain this in detail. In particular we identify clearly that our estimate of EEH is precise but potentially biased high. See section 3.3.

Concerning the dependence of EEH to salinity, when we claim that "that the EEH is most sensitive to salinity" we mean that EEH is relatively more sensitive to salinity than to temperature. However, at global scale, salinity changes are very small, so the EEH changes as well.

In addition, to avoid Argo issues on salinity measurements, we used salinity until 2015 only. It prevents from the recent drifts in the Argo record of salinity (see for example Ponte et al. 2021).

→ The sentence dealing with the EEH uncertainty was completed (section 3.3)

* The authors may be aware that in parallel to this paper a similar one has come out: Hakuba et al 2021 (https://agupubs.onlinelibrary.wiley.com/doi/abs/10.1029/2021GL093624) They actually find a larger EEI (0.9 W/m2). This paper would in fact be a nice opportunity to put these numbers in perspective (e.g. why do they arrive at a higher number?). Since the GIA

correction on GRACE has such a large effect I indeed do wonder whether the GIA corrections can be one of the culprits.

Over a similar period (2005-2015), results obtained by Hakuba et al., 2021 (trends_global.xlsx provided with the article) are in agreement with ours:

- Hakuba et al., 2021: EEI mean: 0.77 W/m² [-0.51 W/m²- 1.05 W/m²] (2005-2015)
- Marti et al., 2021: 0.77+ 0.24 W/m² (January 2005 - December 2015)

However, major differences are noted with regards to the input data. First, the value of the global EEH differs from one article to the other and the impact of this value is strong. Then, GRACE data used in Hakuba et al., 2021 is a mascon solution (JPL) corrected for the GIA using 5000 solutions from Caron et al. 2018. Our spherical harmonic solutions ensemble relies on two different GIA corrections: Caron et al., 2018 and ICE6G-D. This might indeed explain the difference between the ocean mass trends (over 2005-2015, 2.41 mm/yr [1.98 - 2.75] for Hakuba et al. 2021 and 1.80 +/- 0.21 mm/yr for our ensemble solution). It further stresses the fact that estimating uncertainties due to post-processing choices in GRACE solution is necessary to be able to compare different products.

→ The results from Hakuba et al., 2021 are now mentioned in section 6.

**Minor remarks:**

Intro l33 "play a minor role" is the other 10% meant or something else? Yes , it is. The other reservoirs storing 10% of the energy excess have a low contribution to the EEI variations at the timescales of interest.

l72: "innovative algorithms" I don't want to temper your enthousiasm, but maybe "new" or "original" is better here (let the reader judge themselves whether these are innovative). The main novelty of the present work is the implementation of a robust algorithm to propagate the uncertainties.

l89 Neglible Or assumed to be neglible? In the former case maybe provide an estimate from the cited papers.

Several authors indeed conclude that the salinity variations are negligible in the computation of the global mean sea level change, contributing by about 1 % to the global mean sea level change.

-> We have added this order of magnitude and an additional reference.

l172-l173 "It is however .. content" I don't understand this sentence The section was reformulated.

l190 "if included here" -> if it would be allowed to absorb a fraction of the ocean heat uptake Rephrased.

l245: What kind of numeric differencing scheme is used here? Forward differencing scheme (numpy.diff) → We have specified this in the text.

"256: "is implicitly accounted for in the local EEH coefficients". I'm trying to get my head around this, and understand how the salinity effect would be implicitly accounted for. Please clarify (maybe add a formula for to explain this)

The integrated expansion efficiency of heat is computed with the data of 11 Argo products over 2005-2015. (We start in 2005 because the coverage of Argo is global in 2005. We end in 2015 because salinity issues in the Argo products start at the end of 2015 see for example Ponte et al. 2021). We use the mean salinity field of the 11 Argo

products over 2005-2015 to estimate the IEEH (instead of the reference salinity at 35psu). So the IEEH is computed taking into account the density of sea water at the level of the 2005-2015 mean salinity. It means that the density effect (i.e. the halosteric effect) of the mean salinity over 2005-2015 is accounted for in IEEH. However any halosteric effect induced by a change of salinity wrt to the mean over 2005-2015 is not accounted for. So the map, that is computed as a mean over 2005-2015 accounts for salinity but any other map, computed over a different period, would miss a small halosteric contribution from salinity anomaly wrt to the mean salinity over 2005-2015.

→ The explanations were reformulated and completed.

l289-299 "Thus on the overall ... we neglect it here". I don't really understand the word "correlation" in this context, and why it would be an argument not to apply a GIA correction to altimetry. I suggest to address this together with the comment of the other reviewer on geocentric sea level rise

There is a misunderstanding, because the altimetry is actually corrected from GIA as explained in the dedicated section about the sea-level calculation. In this section we only speak about the correlation of the errors of the GIA and ITRF in the altimetry and gravimetry datasets. We explain that we do the sum of the GMSL and GMOM covariance matrices assuming the errors on GMSL and GMOM are independent of each other. We simply discuss that this assumption is also made for the contribution of GIA and ITRF errors in the sea level and ocean mass estimates. We indeed explain that the level of these errors in each dataset (GMSL and GMOM) is not of the same order and therefore are very poorly correlated.

→ In order to clarify this section, we slightly reformulated the sentence in line 289 without changing the main idea of this paragraph.

l409 power plant -> power plants corrected